# VIDEOANCHOR:
# REINFORCING SUBSPACE-STRUCTURED VISUAL CUES FOR COHERENT VISUAL-SPATIAL REASONING

**Zhaozhi Wang**[1,2]    **Tong Zhang**[1†]    **Mingyue Guo**[2]    **Yaowei Wang**[2]    **Qixiang Ye**[1,2]
[1]University of Chinese Academy of Sciences  [2]Peng Cheng Lab

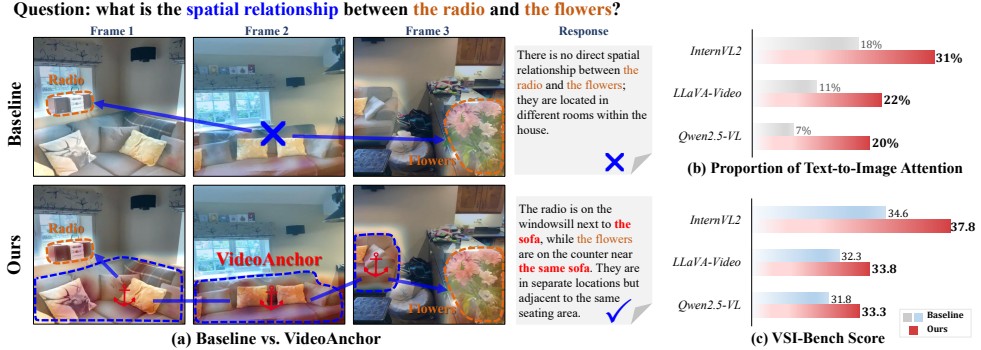

Figure 1: Effect of the proposed VideoAnchor. (a) Attention activations over shared regions (*e.g.*, the sofa) across frames show how consistent patterns are anchored to enhance visual-spatial reasoning, enabling preciser object co-location and contextual proximity than InternVL2-8B (baseline). (b-c) Comparison of text-to-image attention proportion within text-to-all attention and the performance w/o VideoAnchor. Values are averaged across text tokens based on first-layer attention weights.

## ABSTRACT

Multimodal Large Language Models (MLLMs) have achieved impressive progress in vision–language alignment, yet they remain limited in visual–spatial reasoning. We first identify that this limitation arises from the attention mechanism: visual tokens are overshadowed by language tokens, preventing the model from consistently recognizing the same visual cues across frames. To address this challenge, we draw a novel connection between the self-expressiveness property in sparse subspace clustering and the attention mechanism in Transformers. Building on this insight, we propose VideoAnchor, a plug-and-play module that leverages subspace affinities to reinforce visual cues across frames without retraining, effectively anchoring attention to shared visual structures. Extensive experiments across benchmarks and backbone models show consistent performance gains — *e.g.*, 3.2% and 4.6% improvements on VSI-Bench and Video-MME (spatial-related tasks) with InternVL2-8B and Qwen2.5VL-72B—while qualitative analyses demonstrate more coherent subspace partitions and stronger visual grounding. Our codes are available at https://github.com/feufhd/VideoAnchor.

## 1 INTRODUCTION

Advancing *visual-spatial reasoning* — the ability to reason about relative positions, common structures, or spatial references across multiple visual inputs — is crucial for embodied AI, robotics, and spatially grounded decision-making. Despite impressive progress in multimodal large language models (MLLMs) for semantic understanding (Achiam et al., 2023; Liu et al., 2023b; Comanici et al., 2025), current models still exhibit limited capability in this dimension (Yang et al., 2025; Yeh et al.,

---

[†]Correspondence to ✉ tongzhang@ucas.ac.cn.

2025; Ouyang et al., 2025; Li et al., 2025b). This limitation arises primarily from their strong reliance on textual priors, which tends to overshadow fine-grained visual cues and results in hallucinated or visually inconsistent predictions (Ghatkesar et al., 2025). A key reason is that the contributions of individual image tokens are too weak to establish a coherent representational structure. In short, existing MLLMs lack a mechanism to consistently preserve visual cues across frames, which often results in hallucinations and prevents them from achieving coherent visual–spatial reasoning.

This limitation calls for a general mechanism that can reliably preserve visual evidence within attention. Existing designs such as patch-level attention, however, treat tokens independently and thus struggle to capture the higher-level continuity of objects or regions across frames. One promising direction is to organize tokens to semantic subspaces, which directly motivates our approach, VideoAnchor, Fig. 1(a). VideoAnchor leverages this principle to guide attention toward consistent subspace structures, enhancing attention to shared regions across frames and anchoring coherent visual patterns. To avoid the heavy retraining costs and the risk of overfitting in dataset-centric approaches, we instead focus on improving visual–spatial reasoning directly at test time.

Building on this intuition, we propose **VideoAnchor**, a plug-and-play module that enhances visual–spatial reasoning in MLLMs at test time. VideoAnchor establishes a novel connection between the self-expressiveness property in sparse subspace clustering (SSC) (Ji et al., 2017; Zhang et al., 2019) and the attention mechanism in Transformers (Vaswani et al., 2017), and it is composed of two units. First, the Subspace-to-Scaler Unit leverages SSC to organize tokens into semantic subspaces. For each token, it estimates how strongly the token can be expressed by other tokens within the same subspace, thereby reflecting its stability and representativeness in the shared structure. Tokens with stronger expression are regarded as more stable components of coherent visual elements. Building on these estimates, the Attention Regularization Unit incorporates them as scalers into both query–key similarities and value updates, amplifying the influence of tokens that exhibit stronger subspace coherence. Together, these units anchor attention to consistent visual structures across frames, mitigating the dominance of textual priors and ultimately enabling more coherent visual–spatial reasoning, as further illustrated in Fig. 1(b)(c).

The contributions of this study include:

- We introduce VideoAnchor, a plug-and-play module that enhances visual–spatial reasoning in MLLMs at test time without requiring additional training.

- To capture consistent visual anchors across frames, we design a Subspace-to-Scaler Unit that leverages SSC to construct semantic subspaces and generate token-level scalers.

- To reinforce visual grounding within the attention mechanism, we develop an Attention Regularization Unit that integrates the generated scalers into the attention process, amplifying coherent visual tokens.

We extensively evaluate VideoAnchor with different MLLMs on VSI-Bench, All-Angles-Bench, SPAR-Bench, and Video-MME, and demonstrate consistent performance improvements. On VSI-Bench, VideoAnchor achieves a 3.2% improvement when applied to InternVL2-8B (Chen et al., 2024b), while on All-Angles-Bench, it brings a 3.1% gain with LLaVAVideo-72B (Zhang et al., 2025b). Beyond quantitative results, visualizations of clustering outputs show that VideoAnchor produces more semantically coherent subspace partitions over the baseline. Analyses of attention maps further demonstrate that VideoAnchor strengthens attention on visual anchors and mitigates over-reliance on textual priors.

## 2 RELATED WORKS

### 2.1 MULTIMODAL LARGE LANGUAGE MODELS

Multimodal large language models (MLLMs) have emerged as a powerful paradigm for integrating text, images, and other modalities (Yin et al., 2024). Extending the success of large language models (LLMs) (Achiam et al., 2023; Touvron et al., 2023; Team et al., 2024; Guo et al., 2025), they enable the joint synthesis and interpretation of multimodal information. Early efforts such as CLIP (Radford et al., 2021) aligned vision and language through large-scale contrastive learning, while BLIP (Li et al., 2022) and BLIP-2 (Li et al., 2023) advanced vision–language understanding through pre-training

and instruction tuning. Building on this foundation, Flamingo (Alayrac et al., 2022) employed cross-attention to connect visual features with frozen LLMs, enabling few-shot reasoning. More recent systems such as LLaVA (Liu et al., 2023b), Qwen-VL (Bai et al., 2023; Wang et al., 2024; Bai et al., 2025), and InternVL (Chen et al., 2024c;b;a) integrate visual encoders with language backbones, achieving strong performance across tasks such as image captioning, visual question answering, and multi-image reasoning. Despite these advances, MLLMs still face challenges in visual–spatial reasoning. Their strong reliance on textual priors often suppresses fine-grained spatial cues (Ghatkesar et al., 2025), leaving image tokens underrepresented and leading to hallucinated or incoherent predictions when performing visual–spatial reasoning across frames.

## 2.2 VISUAL-ENHANCED MLLMS

To address the insufficient visual grounding in MLLMs, researchers have explored different strategies to strengthen the visual component. One prominent line of work improves high-resolution input (Luo et al., 2025; Guo et al., 2024; Li et al., 2024), which divides images into patches or combines multiple resolutions to capture fine-grained details. Another line of research enhances feature fusion (Lin et al., 2023; Shi et al., 2024; Yao et al., 2024), integrating diverse encoders or deformable attention mechanisms to enrich visual representation, though often at the cost of more tokens and computation.

To overcome the retraining overhead and scalability issues of these approaches, a third line of work investigates training-free solutions. Representative examples include $DC^2$ (Wang et al., 2025), ControlMLLM (Wu et al., 2024), and VisionFuse (Chen et al., 2024d), which enhance perception at inference by recursive image partitioning, attention adjustment, or multi-encoder fusion. However, these approaches still face trade-offs: $DC^2$ incurs inference overhead, ControlMLLM requires manual region specification, and VisionFuse depends on multiple encoders. Thus, enhancing the visual perception of MLLMs in a scalable and adaptive manner remains an open challenge.

## 2.3 VISUAL-SPATIAL REASONING

Early benchmarks (Johnson et al., 2017; Liu et al., 2023a; Tang et al., 2024; Ramakrishnan et al., 2025) primarily assessed this ability in single images, probing spatial relations within static scenes. More recent efforts (Yang et al., 2025; Yeh et al., 2025; Ouyang et al., 2025; Li et al., 2025b) extend the evaluation to videos and multi-view input, which better reflect real-world spatial perception but also reveal persistent weaknesses in current MLLMs.

To improve performance, some approaches (Feng et al., 2025; Wu et al., 2025a; Liao et al., 2025) construct video-centric datasets to enhance spatial awareness, but these methods incur heavy retraining costs and risk limited generalization. In contrast, our proposed VideoAnchor enhances spatial reasoning directly at test time in a training-free, plug-and-play manner, offering efficiency, broad compatibility with existing MLLMs, and reliable generalization without retraining.

## 3 METHODOLOGY

### 3.1 VIDEOANCHOR OVERVIEW

The framework of VideoAnchor is shown in Fig. 2. Given a sequence of frames or multi-view images, a pretrained visual encoder first yields visual tokens; **VideoAnchor** then elevates evidence that is *shared across frames*—serving as visual anchors—to improve visual grounding at inference time. To achieve this, VideoAnchor is composed of two units:

- **Subspace-to-Scaler Unit.** The goal of this unit is to identify anchor tokens and convert their importance into a set of simple, multiplicative scalers. This process unfolds in three steps: discovering the underlying geometric structure of the tokens, calculating a "sharing expression score" for each token based on that structure, and converting these scores into scalers for the attention mechanism.

- **Attention Regularization Unit.** Obtained the scalers from the above, we modulate attention at inference by pairwise gating of query–key interactions after the exponential and per-token amplification of values, thereby allocating more probability mass and representational capacity to visual anchor tokens.

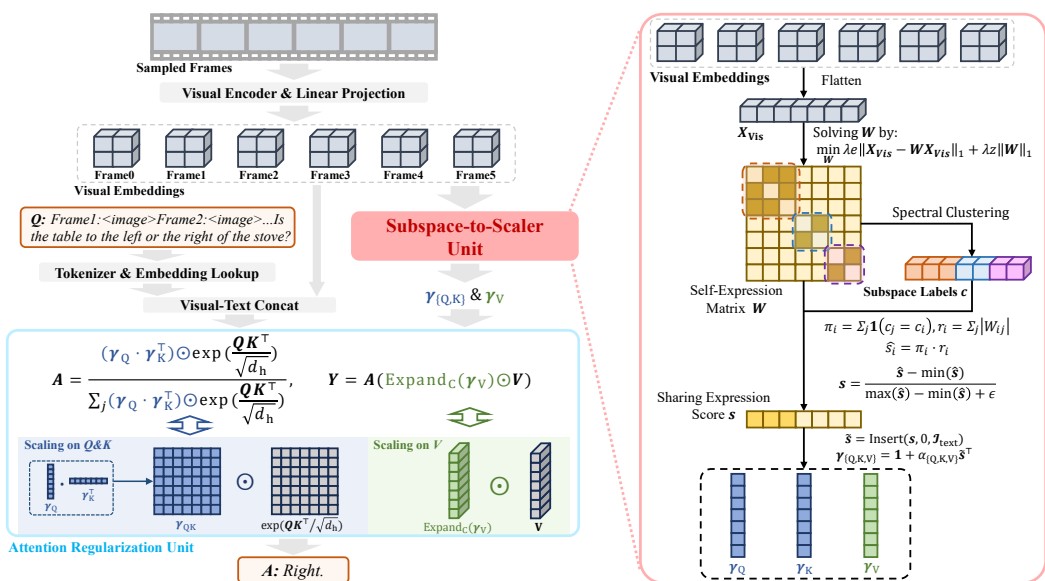

Figure 2: The overall framework of our proposed **VideoAnchor**.

This design improves visual-spatial reasoning without additional training while preserving scalability and generalization.

## 3.2 SUBSPACE-TO-SCALER UNIT

**Self-expressiveness.** The foundation of our approach is the self-expressiveness property, which posits that each data point in a union of subspaces can be represented as a linear combination of other points from the same subspace. We leverage Sparse Subspace Clustering (SSC) to find such a representation. Given the flattened visual tokens $\boldsymbol{X}_{\text{Vis}} \in \mathbb{R}^{N_{\text{Vis}} \times D}$ extracted across frames or views, where $N_{\text{Vis}}$ and $D$ denote the number of visual tokens and the feature dimension respectively, we reveal their underlying subspace structure using Sparse Subspace Clustering (SSC). Specifically, we learn a self-expression matrix $\boldsymbol{W} \in \mathbb{R}^{N_{\text{Vis}} \times N_{\text{Vis}}}$ through

$$\min_{\boldsymbol{W}} \quad \lambda_e \left\| \boldsymbol{X}_{\text{Vis}} - \boldsymbol{X}_{\text{Vis}}\boldsymbol{W} \right\|_1 + \lambda_z \left\| \boldsymbol{W} \right\|_1, \quad \text{s.t. } \text{diag}(\boldsymbol{W}) = 0. \tag{1}$$

Here, $\lambda_e$ enforces reconstruction fidelity under an $\ell_1$ loss, ensuring robustness to noise and perturbations, while $\lambda_z$ imposes sparsity so that each token is reconstructed using only a small number of points from the same union of subspaces. The constraint $\text{diag}(\boldsymbol{W}) = 0$ prevents trivial self-reconstruction and encourages the discovery of meaningful cross-token relations.

Eq. (1) is solved via the Alternating Direction Method of Multipliers (ADMM) (Gabay & Mercier, 1976), which decomposes the problem into iterative updates of the coefficient matrix, its sparse approximation, and the residual. This strategy provides an efficient solver with provable convergence under mild conditions. Details of the update rules are provided in Appendix A.

After optimizing the self-expression matrix $\boldsymbol{W}$, we construct a symmetric adjacency matrix $\boldsymbol{W} + \boldsymbol{W}^\top$ and apply spectral clustering on it to assign subspace labels $c_i$ to each visual token $i$, where each label corresponds to a unique subspace. This process partitions the tokens into distinct semantic groups.

**Sharing expression score.** To identify tokens that can serve as reliable visual anchors, we quantify their "sharing strength" by jointly considering how broadly they are distributed within a subspace (subspace cardinality) and how strongly they contribute to reconstructing others (self-expression row-sum). With the optimized self-expression matrix $\boldsymbol{W}$ and subspace labels $\boldsymbol{c}$, we derive a score for each token that quantifies its importance as a visual anchor. We hypothesize that the best anchors are both structurally representative (i.e., they are important for reconstructing other tokens) and widely

shared (i.e., they belong to a large, coherent semantic cluster. Specifically, for token $i$ we define

$$\pi_i \;=\; \sum_{j=1}^{N} \mathbf{1}(c_j = c_i), \quad r_i \;=\; \sum_{j=1}^{N} |W_{ij}|, \quad \hat{s}_i \;=\; \pi_i \cdot r_i, \tag{2}$$

where $\pi_i$ measures the cardinality of token $i$'s subspace (i.e., how many tokens share the same cluster), and $r_i$ captures the overall reconstruction strength of $i$ in the self-expression matrix $\boldsymbol{W}$. The product $\hat{s}_i$ therefore favors tokens that are not only central to the reconstruction process but also broadly connected within a subspace—intuitively reflecting anchors that are both *structurally representative* and *widely shared*.

Since the absolute scale of $\hat{s}_i$ may vary across samples, we normalize the values to $[0, 1]$ via a min–max adjustment:

$$s_i \;=\; \frac{\hat{s}_i - \min(\hat{\boldsymbol{s}})}{\max(\hat{\boldsymbol{s}}) - \min(\hat{\boldsymbol{s}}) + \epsilon}, \tag{3}$$

with $\epsilon > 0$ ensuring numerical stability. This normalization step facilitates consistent integration across different instances and prevents dominance by outlier tokens. The resulting score vector $\boldsymbol{s} \in \mathbb{R}^{1 \times N_{\mathrm{Vis}}}$ thus encodes the relative anchor strength for all visual tokens.

**Token-wise scalers.** To integrate textual tokens without adding extra emphasis, we extend the visual-only score vector $\boldsymbol{s}$ to length $N$ by assigning a fixed value of $0$ at all text-token positions; here $N = N_{\mathrm{Vis}} + N_{\mathrm{Text}}$ and $N_{\mathrm{Text}}$ denotes the number of text tokens. The resulting $\tilde{\boldsymbol{s}} \in \mathbb{R}^{1 \times N}$ is then converted into multiplicative, token-wise scalers for queries, keys, and values:

$$\boldsymbol{\gamma}_Q \;=\; \mathbf{1} + \alpha_Q \, \tilde{\boldsymbol{s}}^{\top}, \qquad \boldsymbol{\gamma}_K \;=\; \mathbf{1} + \alpha_K \, \tilde{\boldsymbol{s}}^{\top}, \qquad \boldsymbol{\gamma}_V \;=\; \mathbf{1} + \alpha_V \, \tilde{\boldsymbol{s}}^{\top}, \tag{4}$$

where $\alpha_Q, \alpha_K, \alpha_V \geq 0$ control the modulation strength, and $\mathbf{1}$ is an all-ones column vector of length $N$. In our experiments, the values of $\alpha_*$ are set mannually. Note that, when $\alpha_* = 0$, we recover the original, unmodulated attention. With zeros at text positions in $\tilde{\boldsymbol{s}}$, textual tokens receive no additional amplification, whereas higher-scored visual tokens are proportionally upweighted.

### 3.3 ATTENTION REGULARIZATION UNIT

This unit uses the computed scalers $\gamma$ to modify the standard self-attention mechanism, forcing it to prioritize the identified visual anchors. Let per-head queries, keys, and values be $\boldsymbol{Q}, \boldsymbol{K}, \boldsymbol{V} \in \mathbb{R}^{N \times d_h}$, where $d_h$ is the head dimension. We introduce test-time attention regularization, consisting of (i) a pairwise gating mechanism applied to query–key interactions *after* the exponential operation, and (ii) a per-token amplification applied to values. The resulting attention matrix $\boldsymbol{A} \in \mathbb{R}^{N \times N}$ is given by

$$\boldsymbol{A} \;=\; \frac{\left(\boldsymbol{\gamma}_Q \boldsymbol{\gamma}_K^{\top}\right) \odot \exp\!\left(\frac{\boldsymbol{Q}\boldsymbol{K}^{\top}}{\sqrt{d_h}}\right)}{\sum_j \left(\boldsymbol{\gamma}_Q \boldsymbol{\gamma}_K^{\top}\right)_{:,j} \odot \exp\!\left(\frac{\boldsymbol{Q}\boldsymbol{K}^{\top}}{\sqrt{d_h}}\right)_{:,j}}, \tag{5}$$

and the output representation is computed as

$$\boldsymbol{Y} \;=\; \boldsymbol{A}\left(\mathrm{Expand}_c(\boldsymbol{\gamma}_V) \odot \boldsymbol{V}\right). \tag{6}$$

Here $\odot$ denotes element-wise multiplication, and $\mathrm{Expand}_c(\cdot)$ broadcasts the token-wise scaling vector across the channel dimension of $\boldsymbol{V}$.

For practical implementation, instead of modifying the definition of $\mathrm{softmax}(\cdot)$, we adopt a mathematically equivalent formulation that directly incorporates the gating term into the logits:

$$\boldsymbol{A} \;=\; \mathrm{softmax}\!\left(\frac{\boldsymbol{Q}\boldsymbol{K}^{\top}}{\sqrt{d_h}} + \log(\boldsymbol{\gamma}_Q \boldsymbol{\gamma}_K^{\top})\right). \tag{7}$$

This equivalence allows us to compute $\boldsymbol{A}$ without altering the standard softmax implementation, while maintaining the same gating effect as Eq. ( 5).

By applying the gate *after* the exponential, all entries remain nonnegative and the normalization step is preserved, while selectively reweighting attention toward shared anchors [1]. This formulation

---

[1] Please refer to Appendix D for the ablation study of applying the gate before or after the exponential.

Table 1: Performance improvement by VideoAnchor with different MLLMs on VSI-Bench.

| Models | Frames | Avg. | Obj. Count | Abs. Dist. | Obj. Size | Room Size | Rel. Dist. | Rel. Dir. | Route Plan | Appr. Order |
|---|---|---|---|---|---|---|---|---|---|---|
| | | | Numerical Answer | | | | Multiple-Choice Answer | | | |
| InternVL2-2B (Chen et al., 2024b) | 8 | 27.4 | 21.8 | 24.9 | 22.0 | 35.0 | 33.8 | 44.2 | 30.5 | 7.1 |
| * + VideoAnchor | 8 | **28.6 (+1.2)** | 29.0 | 25.4 | 22.0 | 32.5 | 34.6 | 44.6 | 34.0 | 6.5 |
| InternVL2-4B (Chen et al., 2024b) | 8 | 31.7 | 26.3 | 26.9 | 37.5 | 25.4 | 34.5 | 42.9 | 35.0 | 25.4 |
| * + VideoAnchor | 8 | **34.5 (+2.8)** | 43.3 | 29.7 | 33.0 | 26.2 | 35.2 | 44.3 | 40.2 | 23.8 |
| InternVL2-8B (Chen et al., 2024b) | 8 | 34.6 | 23.1 | 28.7 | 48.2 | 39.8 | 36.7 | 30.7 | 29.9 | 39.6 |
| * + VideoAnchor | 8 | **37.8 (+3.2)** | 39.3 | 30.3 | 50.5 | 38.8 | 38.6 | 31.0 | 36.1 | 37.8 |
| Qwen2.5-VL-3B (Bai et al., 2025) | 16 | 26.9 | 18.6 | 16.9 | 16.0 | 26.5 | 39.1 | 45.9 | 29.2 | 23.7 |
| * + VideoAnchor | 16 | **27.8 (+0.9)** | 20.6 | 17.2 | 16.0 | 26.8 | 40.4 | 46.7 | 30.5 | 24.0 |
| Qwen2.5-VL-7B (Bai et al., 2025) | 16 | 31.8 | 28.9 | 15.2 | 45.8 | 30.3 | 37.6 | 40.5 | 25.8 | 30.2 |
| * + VideoAnchor | 16 | **33.3 (+1.5)** | 28.3 | 16.4 | 46.9 | 31.6 | 38.6 | 43.2 | 28.9 | 32.5 |
| LLaVA-Video-7B (Zhang et al., 2025b) | 16 | 34.6 | 44.3 | 14.6 | 45.5 | 25.9 | 41.3 | 40.4 | 36.5 | 29.5 |
| * + VideoAnchor | 16 | **35.7 (+1.1)** | 47.0 | 15.1 | 48.3 | 25.2 | 43.1 | 41.1 | 34.6 | 31.2 |
| LLaVA-Video-72B (Zhang et al., 2025b) | 16 | 39.4 | 43.1 | 23.7 | 54.6 | 38.0 | 41.1 | 35.4 | 30.4 | 49.2 |
| * + VideoAnchor | 16 | **40.9 (+1.5)** | 48.5 | 24.9 | 56.4 | 35.5 | 41.6 | 36.2 | 34.5 | 49.4 |

achieves the same outcome as standard attention while remaining fully compatible with existing Transformer libraries. The final amplification of $V$ by $\gamma_V$ ensures that the information from anchor tokens is not only attended to more strongly but is also given more weight in the final aggregated output representation. In practice, the Attention Regularization Unit is inserted into every block of the MLLM to enhance spatial grounding consistently.

## 3.4 DISCUSSION

• **Why does SSC help visual-spatial reasoning?** SSC groups tokens into semantic subspaces that capture shared visual content across frames. The resulting self-expression matrix $W$ in Eq. (1) can be regarded as a self-supervised form of self-attention (Ji et al., 2017; Zhang et al., 2019; Vidal, 2022), where each token is reconstructed from a sparse set of neighbors within its subspace. This formulation naturally encodes geometric dependencies and higher-level object continuity, providing structural cues that support visual-spatial reasoning tasks.

• **Why does representativeness-aware attention scaling help?** While patch-level attention treats tokens independently and thus often overlooks higher-level semantic continuity, the sharing expression score $s$ in Eq. (2) provides a complementary signal by quantifying token representativeness. It combines subspace cardinality ($\pi_i$) with reconstruction strength ($r_i$), yielding a graded notion of representativeness that reflects both semantic sharing and structural influence. Incorporating $s$ into the attention mechanism biases probability mass toward more representative tokens, ensuring that the aggregated representation emphasizes semantically coherent and structurally central elements. This modulation aligns with the intuition that robust spatial reasoning relies on shared anchors across views or temporal contexts, thereby reinforcing the role of visual evidence in multimodal reasoning.

## 4 EXPERIMENT

To assess the effectiveness of visual–spatial reasoning and the generalization ability of VideoAnchor, we evaluate it on three representative benchmarks—VSI-Bench (Yang et al., 2025) (video inputs) and All-Angles-Bench (Yeh et al., 2025) (multi-view images) for visual–spatial reasoning, and Video-MME (Fu et al., 2025) for general video understanding—using multiple state-of-the-art MLLMs. Detailed settings are provided in Appendix B.

### 4.1 BENCHMARK EVALUATION

**VSI-Bench.** Table 1 reports results on VSI-Bench, which evaluates numerical and multiple-choice visual–spatial reasoning over videos. Across all backbones, VideoAnchor consistently improves performance. For instance, InternVL2-8B (8 frames) gains +16.2 on object counting and +6.2 on

Table 2: Performance improvement by VideoAnchor with different MLLMs on All-Angles-Bench.

| Models | Avg. | Attribute | Cam. Pose | Counting | Manipul. | Rel. Dir. | Rel. Dist. |
|---|---|---|---|---|---|---|---|
| | | Multiple-Choice Answer | | | | | |
| InternVL2.5-2B (Chen et al., 2024a) | 44.2 | 58.7 | 26.7 | 38.6 | 47.8 | 31.8 | 47.5 |
| * + VideoAnchor | **45.5 (+1.3)** | 61.1 | 27.9 | 41.1 | 47.5 | 33.3 | 48.6 |
| InternVL2.5-8B (Chen et al., 2024a) | 48.5 | 77.5 | 26.7 | 48.6 | 39.9 | 34.6 | 51.8 |
| * + VideoAnchor | **50.2 (+1.7)** | 76.3 | 24.3 | 52.4 | 44.0 | 38.2 | 52.5 |
| InternVL2.5-38B (Chen et al., 2024a) | 53.1 | 80.5 | 32.3 | 54.9 | 45.7 | 40.9 | 54.0 |
| * + VideoAnchor | **55.0 (+1.9)** | 82.3 | 33.0 | 57.0 | 48.6 | 43.8 | 54.7 |
| LLaVA-OneVision-0.5B (Li et al., 2025a) | 42.4 | 51.4 | 35.2 | 25.1 | 49.4 | 34.0 | 45.9 |
| * + VideoAnchor | **43.9 (+1.5)** | 55.4 | 36.4 | 26.0 | 49.6 | 37.3 | 46.2 |
| LLaVA-OneVision-7B (Li et al., 2025a) | 44.1 | 63.7 | 16.5 | 37.0 | 42.4 | 35.8 | 50.0 |
| * + VideoAnchor | **46.7 (+2.6)** | 63.3 | 23.9 | 40.6 | 46.0 | 37.7 | 52.0 |
| LLaVA-Video-7B (Zhang et al., 2025b) | 43.1 | 65.5 | 14.8 | 37.4 | 42.6 | 31.5 | 47.3 |
| * + VideoAnchor | **44.3 (+1.2)** | 65.8 | 16.5 | 41.0 | 40.1 | 34.1 | 50.4 |
| LLaVA-Video-72B (Zhang et al., 2025b) | 49.9 | 75.7 | 29.0 | 40.6 | 44.5 | 42.3 | 52.8 |
| * + VideoAnchor | **53.0 (+3.1)** | 77.1 | 33.0 | 41.9 | 45.8 | 44.1 | 60.2 |

room planning, while Qwen2.5-VL-7B shows broad improvements across nearly all categories, including relative direction (+2.7 with 16 frames). Larger models also benefit: LLaVA-Video-72B achieves an overall +1.5 improvement under 16-frame input, with noticeable gains on both numerical and multiple-choice tasks. Interestingly, the improvements tend to scale with model size—smaller backbones such as InternVL2-2B and LLaVA-Video-7B obtain modest gains (+1.2/+1.1 overall), while larger backbones such as InternVL2-8B and LLaVA-Video-72B achieve more substantial overall boosts (+3.2 and +1.5, respectively). These results demonstrate not only the effectiveness of VideoAnchor for video-based visual–spatial reasoning, but also its favorable scaling behavior as model capacity increases.

**All-Angles-Bench.** Table 2 summarizes results on All-Angles-Bench, which evaluates spatial reasoning from multi-view images. Similar to VSI-Bench, VideoAnchor consistently improves performance across backbones. Taking LLaVA-OneVision as an example, the overall gain increases from +1.5 on the 0.5B model to +2.6 on the 7B model, while for LLaVA-Video, the improvement grows from +1.2 on the 7B model to +3.1 on the 72B model. These results suggest a scaling trend, indicating that larger-capacity models may be able to leverage VideoAnchor more effectively and yield stronger improvements. Beyond overall accuracy, VideoAnchor also brings targeted gains in challenging categories, averaging +2.6 on counting and +2.5 on relative direction across the listed models. The improvements further extend to camera pose, manipulation, and relative distance, demonstrating that VideoAnchor generalizes well beyond temporal reasoning to strengthen multi-view visual–spatial consistency.

Table 3: Left: Results on SPAR-Bench. Right: Results on Video-MME (Spatial Perception & Spatial Reasoning).

| Models | Avg. |
|---|---|
| InternVL2-2B | 28.0 |
| * + VideoAnchor | **29.6 (+1.6)** |
| InternVL2.5-2B | 29.9 |
| * + VideoAnchor | **31.2 (+1.3)** |
| InternVL2-4B | 32.0 |
| * + VideoAnchor | **33.4 (+1.4)** |
| InternVL2.5-4B | 30.5 |
| * + VideoAnchor | **33.1 (+2.6)** |

| Models | Frames | Avg. | Spatial Perception | Spatial Reasoning |
|---|---|---|---|---|
| InternVL2.5-8B | 8 | 69.9 | 63.0 | 76.8 |
| * + VideoAnchor | 8 | **74.4 (+4.5)** | 64.8 | 83.9 |
| LLaVA-Video-7B | 16 | 71.6 | 59.3 | 83.9 |
| * + VideoAnchor | 16 | **73.4 (+1.8)** | 61.1 | 85.7 |
| Qwen2.5VL-72B | 16 | 75.4 | 72.2 | 78.6 |
| * + VideoAnchor | 16 | **80.0 (+4.6)** | 77.8 | 82.1 |

**SPAR-Bench.** To validate the generalizability of VideoAnchor across diverse spatial reasoning tasks, we additionally evaluate it on SPAR-Bench (Zhang et al., 2025a), a more comprehensive benchmark covering both single-view and multi-view spatial understanding. As shown in Table 3 (left), VideoAnchor consistently improves the average score by +1.3 to +2.6, demonstrating the generalization and the robustness of VideoAnchor on challenging spatial tasks.

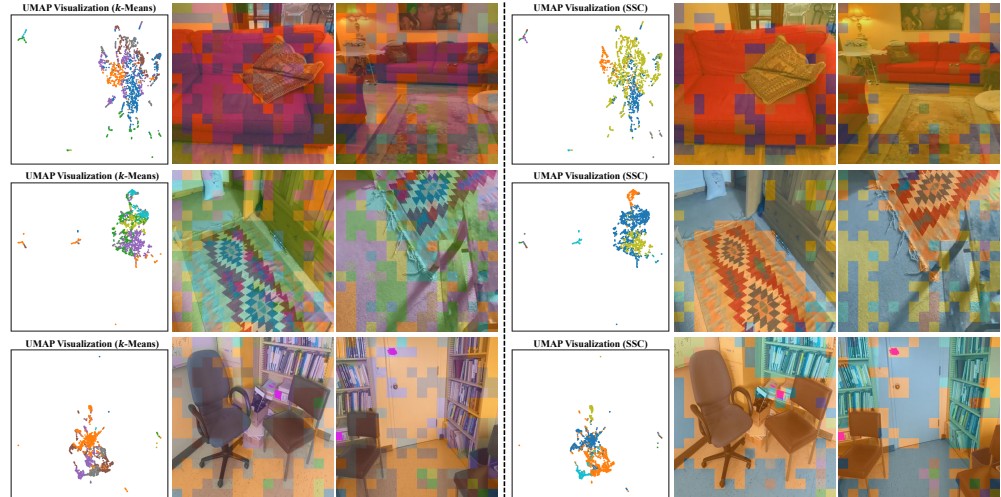

Figure 3: Comparison of visual token embedding clustering using $k$-Means (left) and Sparse Subspace Clustering (SSC, right). Each row shows one scene, with UMAP projections of visual token embeddings (leftmost column) and corresponding token overlay visualizations on two representative frames (middle and right). Compared to $k$-Means, SSC yields purer clusters and better captures cross-frame semantic consistency, effectively grouping spatially aligned regions (e.g., sofa, carpet, chairs) across different views.

**Video-MME.** We further evaluate VideoAnchor on the general-purpose Video-MME benchmark, which spans diverse video understanding tasks. Here we report results on those most relevant to spatial understanding. As shown in Table 3 (right), VideoAnchor yields consistent improvements across models: Qwen2.5VL-72B improves from 75.4 to 80.0 (+4.6) on average, with notable gains in spatial perception (+5.6) and reasoning (+3.5). InternVL2.5-8B and LLaVA-Video-7B achieve +4.5/+1.8 gains, respectively, confirming the generality of VideoAnchor across models and scales. These results show that VideoAnchor enhances both fine-grained spatial perception and higher-level spatial reasoning. Full results on Video-MME are provided in Appendix C.

## 4.2 ANALYSIS

**Subspace bridging and clustering quality.** Fig. 3 illustrates that SSC groups semantically related regions (e.g., sofa, carpet, chairs) into stable clusters that persist across video frames, providing coherent anchors for modeling spatial relations and shared object references. Compared with $k$-Means, which often yields scattered or fragmented clusters due to its reliance on Euclidean distance in high-dimensional space, SSC produces clearer semantic boundaries and more coherent token groupings. This advantage is further confirmed quantitatively in Table 4, where SSC consistently surpasses $k$-Means across categories, achieving a higher overall score on VSI-Bench (37.8 vs. 35.2). Together, these results highlight SSC's ability to form semantically meaningful subspaces that not only persist across frames but also translate into tangible gains in visual-spatial reasoning performance.

Table 4: VSI-Bench performance of VideoAnchor with InternVL2-8B (8 frames) under different attention scaling paradigms: uniform, centroid-based ($k$-Means), and subspace-based (SSC).

| Methods | Baseline | UniformBoost | $k$-Means | **SSC** |
|---------|----------|--------------|-----------|---------|
| Avg. | 34.6 | 34.2 (-0.4) | 35.2 (+0.6) | **37.8 (+3.2)** |

**Cluster to attention enhancement.** Fig. 4 presents the visualizations of the self-expression row-sum, SSC labels, sharing expression score, text-to-image attention maps, and model responses. The self-expression row-sum quantifies each token's overall reconstruction strength when representing others, though small categories may introduce noise. In conjunction with SSC labels that delineate semantically coherent regions across views (*e.g.*, the fireplace and its surrounding structures), it enables the computation of the sharing expression score, which serves as a more robust anchor signal.

**Question: If I'm standing by the TV and facing the entrance, is the fireplace to my left, right, or back?**

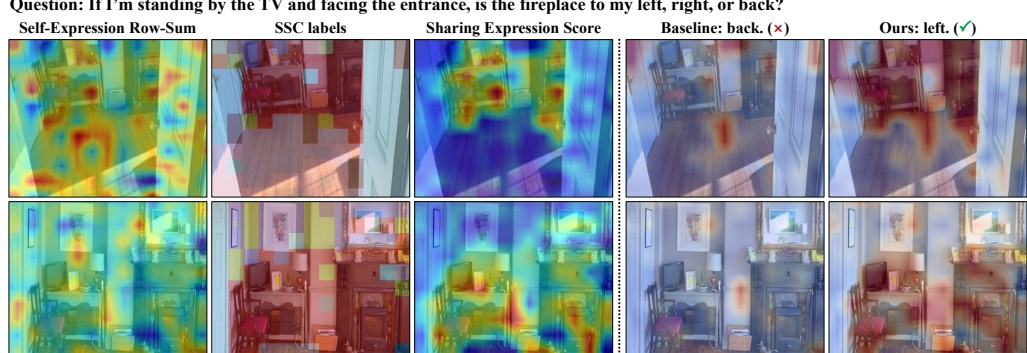

Figure 4: Example question with visualizations of self-expression row-sum, SSC labels, sharing expression score, and text-to-image attention maps, together with responses from the baseline (InternVL2-8B) and our VideoAnchor. SSC-guided attention enhancement improves visual–spatial reasoning and yields the correct answer.

These anchors guide VideoAnchor to reweight tokens and refine the original attention, resulting in sharper and more consistent text-to-image alignments compared with the baseline (InternVL2-8B). Consequently, VideoAnchor attains more accurate visual–spatial reasoning and produces correct responses, whereas the baseline tends to disperse attention and generate errors.

**Effect of scaling positions** ($Q/K/V$) **of VideoAnchor.** Table 5 shows the impact of applying scaling to different attention components. Scaling only on V yields the largest gain (+1.9), while adding Q and V further improves performance (+2.2), with the best result from scaling all $Q/K/V$ (+2.8). This suggests that value scaling primarily enhances visual evidence, while query and key scaling provide complementary benefits that make spatial reasoning more consistent and stable.

Table 5: Ablation of Q/K/V scaling in VideoAnchor on VSI-Bench (InternVL2-4B, 8 frames).

| Methods | Baseline | Scaling on V | Scaling on Q&V | **Scaling on Q&K&V** |
|---------|----------|--------------|----------------|----------------------|
| Avg. | 31.7 | 33.6 (+1.9) | 33.9 (+2.2) | **34.5 (+2.8)** |

**Ablation of the Sharing Expression Score.** Table 6 presents the ablation results on the computation of the sharing expression score. Using only the cluster number results in a slight improvement (+1.0), while relying solely on self-expression coefficients yields results close to the baseline (+0.4). Combining both yields the highest improvement (+3.2), highlighting their complementarity: self-expression characterizes token-level reconstruction strength, whereas cluster number reflects the stability and representativeness of subspaces. Small clusters may introduce noise, and large clusters may dilute distinctiveness; their joint consideration balances these effects and provides a more reliable guidance signal. This demonstrates that the sharing expression score not only stabilizes anchor selection but also enhances the consistency of attention modulation across views, thereby reinforcing visual–spatial reasoning.

Table 6: Ablation of sharing expression score computation in VideoAnchor on VSI-Bench.

| Methods | InternVL2-8B (8 frames) | Cluster number | Self-expression | Cluster number & Self-expression |
|---------|-------------------------|----------------|-----------------|----------------------------------|
| Avg. | 34.6 | 35.6 (+1.0) | 35.0 (+0.4) | **37.8 (+3.2)** |

**Robustness to frame sampling.** Table 7 reports results on VSI-Bench with different frame sampling settings. VideoAnchor consistently improves performance across 8/16/32/64 frames for both InternVL2-8B and Qwen2.5-VL-7B. These results indicate that the benefits of VideoAnchor are stable under varying numbers of input frames, demonstrating robustness to frame sampling strategies. Compared to Video-R1 (Feng et al., 2025), a GRPO-fine-tuned Qwen2.5-VL model, our method maintains robustness across different numbers of input frames, whereas Video-R1 surpasses us only under the specific setting it was fine-tuned on. Please refer to Appendix F for results and further analysis.

Table 7: VSI-Bench performance with VideoAnchor under different frame sampling settings.

| Sampling Frames | 8 | 16 | 32 | 64 | Sampling Frames | 8 | 16 | 32 |
|---|---|---|---|---|---|---|---|---|
| InternVL2-8B | 34.6 | 36.8 | 37.4 | 37.6 | Qwen2.5VL-7B | 28.3 | 31.8 | 36.4 |
| * + VideoAnchor | 37.8 (+3.2) | 38.1 (+1.3) | 39.2 (+1.8) | 39.8 (+2.2) | * + VideoAnchor | 29.5 (+1.2) | 33.3 (+1.5) | 37.4 (+1.0) |

**Sensitivity analysis of $\alpha_{Q/K/V}$.** To further validate the robustness in $\alpha_{Q/K/V}$ setting of VideoAnchor, we conduct experiments with InternVL2-4B (8 frames) on VSI-Bench. As detailed in Table 8, we varied $\alpha_Q$, $\alpha_K$, and $\alpha_V$ with a step size of 0.2 within the ranges of $[1.5, 3.5]$, $[1.0, 3.0]$, and $[2.0, 4.0]$, respectively. The results suggest that the model performance remains relatively consistent across these variations, with scores fluctuating within the range of 33.6 to 34.5. Furthermore, these configurations consistently show improvements over the baseline score of 31.7, implying that VideoAnchor may not be overly sensitive to precise hyperparameter tuning.

Table 8: Sensitivity analysis of $\alpha_{Q/K/V}$ on VSI-Bench with InternVL2-4B (8 frames).

| Tuning Parameters | Values | VSI-Bench Score |
|---|---|---|
| Baseline | - | 31.7 |
| $\alpha_Q$ | 1.5-3.5 | 34.3-34.5 (+2.6-2.8) |
| $\alpha_K$ | 1.0-3.0 | 34.0-34.5 (+2.3-2.8) |
| $\alpha_V$ | 2.0-4.0 | 33.6-34.5 (+1.9-2.8) |

**Runtime comparison.** In VideoAnchor, the iterative ADMM optimization in SSC introduces additional latency and additional complexity ($O(N^2)$ with the number of visual tokens $N$) compared to the baseline. However, to mitigate this, we implemented the ADMM solver using CuPy for efficient GPU acceleration, which significantly reduces the computational bottleneck. Furthermore, we notice that VideoAnchor is robust to the ADMM convergence threshold ($\epsilon$), thus choosing suitable relaxed thresholds reduces runtime with marginal impact on the performance. Measured on NVIDIA V100 GPUs, the optimized VideoAnchor increases inference latency by only 0.8s per sample while yielding a substantial +3.2 gain on VSI-Bench (as shown in Table 9), demonstrating a highly acceptable trade-off between efficiency and accuracy.

Table 9: The runtime and the performance comparisons of using VideoAnchor on VSI-Bench with InternVL2-8B (8 frames).

| Models | Runtime (s/iter) | VSI-Bench Score |
|---|---|---|
| InternVL2-8B (8 frames) | 6.0 | 34.6 |
| * + VideoAnchor | 6.8 | 37.8 (+3.2) |

## 5 CONCLUSIONS

In this work, we presented **VideoAnchor**, a plug-and-play module that enhances visual–spatial reasoning in multimodal large language models at test time. We identified that MLLMs' reasoning failures often stem from an inability to ground themselves in consistent visual evidence across frames. To address this, VideoAnchor leverages the self-expressiveness property from sparse subspace clustering to identify and amplify shared visual structures, effectively anchoring the model's attention without requiring any retraining. Across visual-spatial reasoning benchmarks, VideoAnchor achieves consistent improvements from lightweight to large-scale models, validating our core hypothesis and highlights test-time attention regularization based on semantic subspaces as a promising and efficient direction for improving the visual–spatial reasoning.

## 6 ACKNOWLEDGMENT

This work was supported by National Natural Science Foundation of China (NSFC) under Grant 62225208 and 62521007, and CAS Project for Young Scientists in Basic Research under Grant No.YSBR-117.

**Reproducibility Statement.** The source code implementing VideoAnchor has been submitted in the supplementary material. Experimental settings and part of the hyperparameters are provided in the main text and the appendix to facilitate the reproducibility of our results.

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

# A  ADMM SOLVER FOR SSC

To solve Eq. 1, we introduce auxiliary variables and apply the Alternating Direction Method of Multipliers (ADMM). The optimization problem is decomposed into three main steps at each iteration:

- **Update of coefficient matrix $\boldsymbol{A}$:**

$$\boldsymbol{A} \leftarrow \left(\lambda_z\,\boldsymbol{X}_{\mathrm{Vis}}^{\top}\boldsymbol{X}_{\mathrm{Vis}} + \rho\boldsymbol{I} + \rho\mathbf{1}\mathbf{1}^{\top}\right)^{-1}\left(\lambda_z\,\boldsymbol{X}_{\mathrm{Vis}}^{\top}(\boldsymbol{X}_{\mathrm{Vis}} - \boldsymbol{E}) + \rho(\boldsymbol{W}_{iter} + \mathbf{1}\mathbf{1}^{\top}) - \mathbf{1}\Delta_1^{\top} - \Delta_2\right), \quad (8)$$

  where $\rho$ is the penalty parameter and $\Delta_1, \Delta_2$ are dual variables.

- **Update of sparse representation $\boldsymbol{W}_{iter}$ via soft-thresholding:**

$$\boldsymbol{W}_{iter} \leftarrow \mathcal{S}_{1/\rho}\!\left(\boldsymbol{A} + \tfrac{1}{\rho}\Delta_2\right), \quad \boldsymbol{W}_{iter} \leftarrow \boldsymbol{W}_{iter} - \mathrm{diag}(\boldsymbol{W}_{iter}), \quad (9)$$

  where $\mathcal{S}_\eta(v) = \mathrm{sign}(v)\,\max(|v|-\eta, 0)$ denotes the element-wise soft-thresholding operator.

- **Update of residual $\boldsymbol{E}$:**

$$\boldsymbol{E} \leftarrow \mathcal{S}_{\lambda_e/\lambda_z}\!\left(\boldsymbol{X}_{\mathrm{Vis}} - \boldsymbol{X}_{\mathrm{Vis}}\boldsymbol{A}\right). \quad (10)$$

- **Dual updates:**

$$\Delta_1 \leftarrow \Delta_1 + \rho(\boldsymbol{A}^{\top}\mathbf{1} - \mathbf{1}), \quad (11)$$

$$\Delta_2 \leftarrow \Delta_2 + \rho(\boldsymbol{A} - \boldsymbol{W}_{iter}). \quad (12)$$

The iterations are repeated until convergence, typically checked via $\|\boldsymbol{A} - \boldsymbol{W}_{iter}\|_\infty < \epsilon$. The final $\boldsymbol{W}_{iter}$ is used as the self-expression matrix $\boldsymbol{W}$.

# B  DETAILED SETTINGS

For the Sparse Subspace Clustering (SSC) module, we adopt the ADMM solver described in Appendix A. For most experiments, the penalty parameter $\rho$ is set to 300, and the sparsity weight $\lambda_z$ is set to 800, while the reconstruction weight $\lambda_e$ is set to 800. The convergence tolerance $\epsilon$ is chosen as $2e^{-4}$, and the maximum number of iterations is capped at 10000.

After obtaining the self-expression matrix $\boldsymbol{W}$, we apply column-wise thresholding to retain only the most significant reconstruction coefficients, controlled by a threshold parameter $threshold\_c$ (set to 1 in our experiments unless otherwise specified). We then construct an affinity matrix by symmetrization $\boldsymbol{W} + \boldsymbol{W}^{\top}$. Spectral clustering is finally applied with the number of subspaces fixed to 24. In our implementation, we set the nearest-neighbor parameter $K = 0$, meaning that all edges are retained when building the similarity graph, and clustering is solved via normalized cuts.

In our VSI-Bench implementation, the adopted values of $\alpha_Q, \alpha_K, \alpha_V$ for representative MLLMs are as reported in Table 10. We also observe that moderate variations in these values do not substantially affect the performance gains, suggesting the robustness of VideoAnchor to this choice.

In all experiments, we adopt $do\_sample = False, num_beams = 1$ during inference. Under this setting, sampling-related parameters such as temperature and top-p are not used for generation and thus have no effect on the output. Consequently, the models' responses are fully deterministic and consistent across runs, eliminating the possibility of inference variance.

Table 10: Scaling coefficients ($\alpha_Q$, $\alpha_K$, $\alpha_V$) for representative MLLMs in VideoAnchor on VSI-Bench.

| Models | $\alpha_Q$ | $\alpha_K$ | $\alpha_V$ |
|---|---|---|---|
| InternVL2-4B | 2.5 | 2.0 | 3.0 |
| InternVL2-8B | 4.0 | 9.5 | 2.5 |
| LLaVA-Video-7B | 2.0 | 2.0 | 0.5 |
| Qwen2.5VL-7B | 3.5 | 3.5 | 0.25 |

## C  VIDEO-MME RESULTS

Table 11 presents results on the Video-MME benchmark. VideoAnchor consistently improves performance across all evaluated models, with gains ranging from +0.3 to +1.2. The improvements are especially notable on spatial-related tasks in Table 3 (right), confirming that VideoAnchor is primarily designed to strengthen visual–spatial perception and reasoning.

Table 11: Results on Video-MME.

| Models | Frames | Avg. | Short | Medium | Long |
|---|---|---|---|---|---|
| InternVL2-4B | 8 | 49.8 | 60.0 | 47.2 | 42.2 |
| * + VideoAnchor | 8 | **51.0 (+1.2)** | 61.6 | 48.2 | 43.1 |
| InternVL2.5-8B | 8 | 57.9 | 68.1 | 56.6 | 49.1 |
| * + VideoAnchor | 8 | **58.2 (+0.3)** | 68.2 | 56.7 | 49.6 |
| LLaVA-Video-7B | 16 | 59.9 | 70.9 | 58.7 | 50.1 |
| * + VideoAnchor | 16 | **60.4 (+0.5)** | 71.3 | 59.4 | 50.3 |
| Qwen2.5VL-72B | 16 | 63.5 | 71.8 | 61.8 | 56.9 |
| * + VideoAnchor | 16 | **63.8 (+0.3)** | 72.4 | 62.1 | 57.1 |

## D  SCALING Q&K BEFORE VS. AFTER SOFTMAX

Table 12 compares the effect of applying Q&K scaling before versus after the Softmax operation. Applying the scaling before Softmax severely degrades performance (5.2, –29.4), as it distorts the similarity scores and destabilizes the resulting attention weights. In contrast, applying the scaling after Softmax yields a clear improvement over the baseline (+3.2), confirming the effectiveness of our design. By introducing the gate after the exponential, all entries remain nonnegative and the normalization step is preserved, while attention can be selectively reweighted toward shared anchors. This design ensures that VideoAnchor emphasizes spatially coherent tokens without disrupting the overall distribution of attention weights.

Table 12: Ablation study on applying Q&K scaling before vs. after the Softmax operation in VideoAnchor on VSI-Bench.

| Methods | InternVL2-8B (8 frame) | Before softmax$(\cdot)$ | After softmax$(\cdot)$ |
|---|---|---|---|
| Avg. | 34.6 | 5.2 (-29.4) | **37.8 (+3.2)** |

## E  EFFECT OF SUBSPACE AND BOOSTED SUBSPACE COUNTS

Table 13 studies the role of subspace numbers and boosted subspace counts. On the left, performance increases as the number of subspaces grows, reaching the best result at 24 (37.8), which suggests that 24 subspaces provide the most suitable semantic partitioning for visual–spatial reasoning in this scenario. Intuitively, using too few subspaces may merge distinct visual patterns together, while too many can fragment coherent regions; around 24 appears to strike a balance between these extremes. On the right, boosting all subspaces consistently achieves the highest score, whereas partially boosting them leads to slight drops, indicating the importance of enhancing every subspace rather than a subset.

Table 13: Ablation study on the number of subspaces (left) and boosted subspaces (right) on VSI-Bench with InternVL2-8B (8 frames). In the left table, all subspaces are boosted by default, and performance is reported as the number of subspaces varies. In the right table, the total number of subspaces is fixed at 24, and only a subset is boosted, chosen in descending order of token frequency.

| Subspace | 12 | 16 | 20 | 24 | 28 | 32 | 36 |
|---|---|---|---|---|---|---|---|
| Average | 36.2 | 36.8 | 37.3 | **37.8** | 36.8 | 37.3 | 36.7 |

| Boosted Subspace | 24 | 22 | 20 | 18 | 16 |
|---|---|---|---|---|---|
| Average | **37.8** | 37.6 | 37.3 | 37.4 | 37.2 |

## F    COMPARISON WITH GRPO-TRAINED MLLMS

Table 14 compares VideoAnchor and Video-R1-7B, both based on the Qwen2.5-VL-7B backbone. With 16 input frames, Video-R1 achieves a higher score (34.6 vs. 33.3), reflecting the advantage of its GRPO-based training tailored to this setting. However, when the number of frames increases, VideoAnchor demonstrates stronger generalization: at 32 frames it clearly surpasses Video-R1 (37.4 vs. 35.8), even slightly outperforming the 64-frame variant of Video-R1 (37.4 vs. 37.1). A plausible explanation is that Video-R1 is optimized during training with a fixed 16-frame configuration, which limits its adaptability to other sampling strategies. In contrast, VideoAnchor avoids such overfitting by directly reinforcing spatially consistent cues at test time, thereby offering a more robust and efficient solution for enhancing visual–spatial reasoning under varying frame settings.

Table 14: Video-R1-7B (GRPO-trained with Qwen2.5-VL) vs. VideoAnchor (test-time).

| Models | Frames | VSI-Bench Score |
|---|---|---|
| Qwen2.5-VL | 16 | 31.8 |
| Qwen2.5-VL + VideoAnchor | 16 | 33.3 |
| Video-R1 | 16 | 34.6 |
| Video-R1 + VideoAnchor | 16 | **36.2** |
| Qwen2.5-VL | 32 | 36.4 |
| Video-R1 | 32 | 35.8 |
| Video-R1 | 64 | 37.1 |
| Qwen2.5-VL + VideoAnchor | 32 | **37.4** |

## G    TEXT-TO-IMAGE ATTENTION COMPARISON

As shown in Fig. 5, VideoAnchor enhances attention more effectively to semantically relevant regions compared to the baseline. In particular, it strengthens focus on objects such as the cabinet and shoes, leading to sharper and more coherent visual grounding.

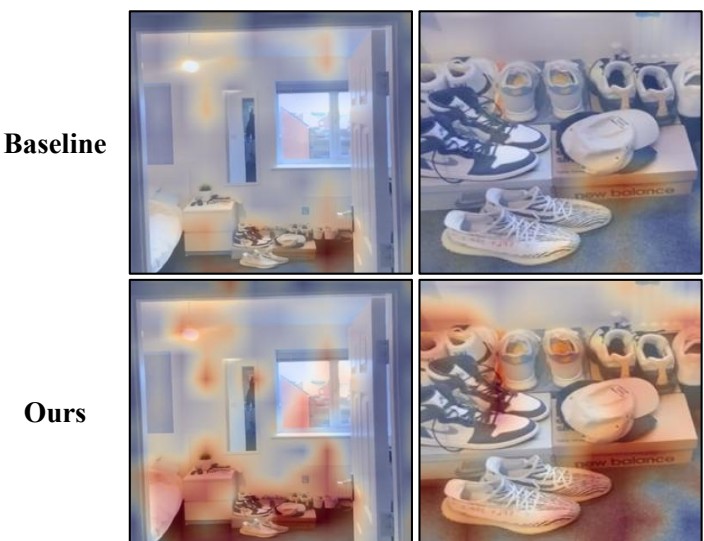

Figure 5: Text-to-image attention comparison of InternVL2-8B (baseline) and with VideoAnchor (ours). Values are averaged across text tokens based on first-layer attention weights.

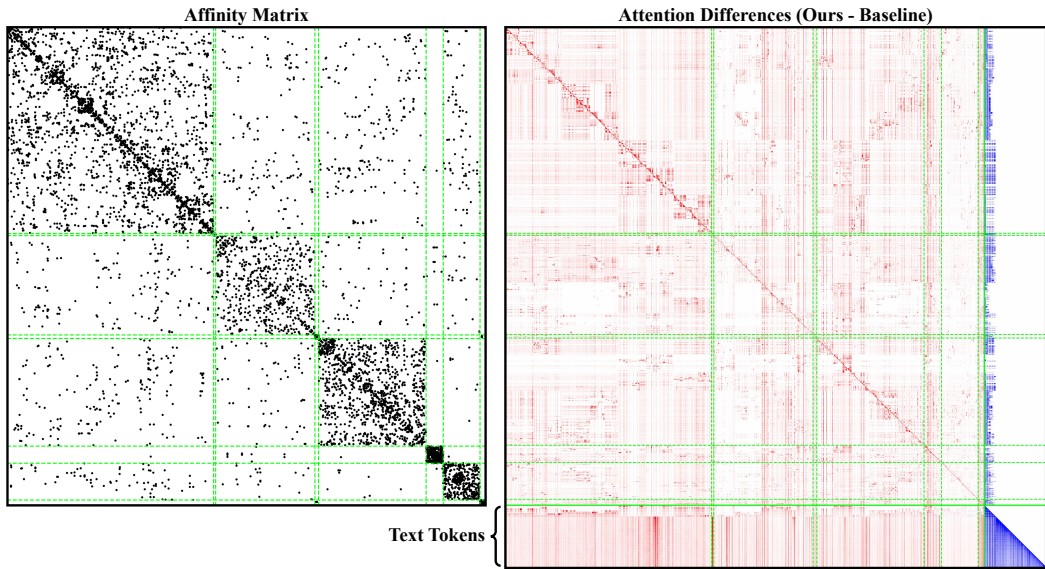

Figure 6: Visualization of subspace structures and attention modulation. Left: Affinity matrix of visual tokens from InternVL2-8B, showing 10 clustered subspaces. Right: Attention differences between VideoAnchor and the baseline (red = increase, blue = decrease). Tokens are reordered with visual tokens (grouped by subspace) preceding text tokens; green dashed lines mark subspace boundaries.

## H  SUBSPACE-AWARE ATTENTION MODULATION

As shown in Fig. 6, VideoAnchor substantially reduces self-attention among text tokens while consistently enhancing attention towards visual tokens. This shift indicates that the model allocates less capacity to redundant linguistic priors and instead emphasizes visual grounding. Moreover, the enhancement is not uniformly distributed: attention gains are concentrated along subspace boundaries, suggesting that VideoAnchor strengthens interactions both within and across subspaces. This demonstrates that subspace-aware modulation effectively bridges semantic structures in the visual domain while mitigating the dominance of textual bias.

## I  COMPATIBILITY WITH FLASHBIAS

VideoAnchor introduces a multiplicative re-weighting of attention, which can be equivalently expressed as an additive log-bias added before the softmax, as provided by Eq. 7. In principle, such a bias could be injected into FlashAttention by converting it into a dense $attention\_mask$. However, this approach leads to prohibitively high memory consumption, especially for long video sequences.

To address this, we refer to recent proposed FlashBias (Wu et al., 2025b), a high-performance inference framework designed to integrate prior-driven attention biases compatible with efficient backends like Triton or PyTorch's SDPA. Given a modified attention

$$\boldsymbol{O} = softmax(\frac{\boldsymbol{Q}\boldsymbol{K}^{\top}}{d_h} + \boldsymbol{B})\boldsymbol{V}$$

, when the attention bias is low-rank and can be decomposed as $\boldsymbol{B} = \boldsymbol{Q}_{bias} * \boldsymbol{K}_{bias}^{T}$, FlashBias provides a kernel implementation compatible with FlashAttention2—without any dense mask construction—and the inference cost becomes essentially identical to pure FlashAttention (no extra memory overhead, no kernel modification).

The introduced $attention\_bias$ in VideoAnchor is $\log(\boldsymbol{\gamma}_Q \boldsymbol{\gamma}_K^{\top})$, which can be exactly decomposed with $rank = 2$ by:

$$\log(\boldsymbol{\gamma}_Q \boldsymbol{\gamma}_K^{\top}) = [\log(\boldsymbol{\gamma}_Q), \mathbf{1}_{n_Q \times 1}][\mathbf{1}_{n_K \times 1}, \log(\boldsymbol{\gamma_K})]^{\top}.$$

This allows us to directly adopt the FlashBias kernel and guarantees: no need for dense masking, no extra memory overhead, and the inference efficiency that could match standard FlashAttention. Thus, VideoAnchor naturally fits into the class of attention-bias mechanisms supported efficiently by FlashBias, providing both theoretical correctness and practical compatibility with modern high-performance inference frameworks. With FlashBias, we test VideoAnchor's inference efficiency based on Qwen2.5-VL-7B (16 frames) on VSI-Bench using NVIDIA 80G A100 GPUs. Note that for this kernel-level comparison, the SSC clustering results were pre-cached. The runtime, GPU memory allocation, and the results are provided in Table 15. VideoAnchor incurs negligible latency increase and maintains the GPU memory consumption compared to the baseline, validating that attention regularization units in VideoAnchor are compatible with high-performance inference frameworks.

Table 15: The runtime / GPU memory / VSI-Bench performance of VideoAnchor with FlashBias.

| Models | Runtime (s/iter) | GPU Memory (GB) | VSI-Bench Score |
|---|---|---|---|
| Qwen2.5-VL-7B (16 frames, FlashAttention) | 5.8 | 38.7 | 31.8 |
| * + VideoAnchor (16 frames, FlashBias-SDPA) | 6.1 | 38.7 | 33.3 (+1.5) |

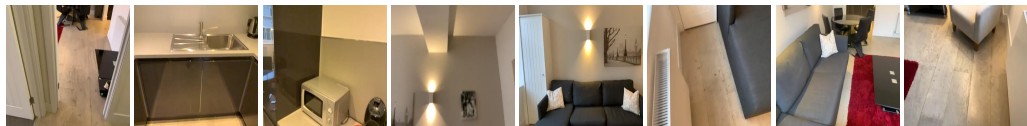

Figure 7: Failure Case 1. Question: How many sofa(s) are in this room? GT: 3. Baseline response: 1. VideoAnchor response: 2.

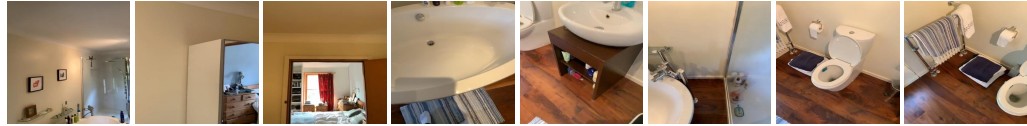

Figure 8: Failure Case 2. Question: If I am standing by the bathtub and facing the bed, is the table to my left, right, or back? An object is to my back if I would have to turn at least 135 degrees in order to face it. GT: Left. Baseline & VideoAnchor: Back.

## J    FAILURE CASES

While VideoAnchor demonstrates significant improvements in visual-spatial reasoning, it still has some limitations. To better understand the generalization boundaries of VideoAnchor, we analyze two representative failure cases in VSI-Bench with InternVL2-8B (8 frames). In failure case 1, as shown in Fig. 7, the model may exhibit counting ambiguities; this stems from the intrinsic mechanism of Sparse Subspace Clustering (SSC), which tends to merge highly correlated visual features into a single subspace, thereby blurring the attention boundaries between distinct but near-identical instances. In failure case 2, as shown in Fig. 8, the model struggles with questions regarding fleeting events that occur between sampled frames. This limitation is attributed to the discrete nature of temporal sampling rather than the clustering module itself, as VideoAnchor functions as a feature enhancement mechanism and cannot recover visual information physically absent from the input grid.

## K    COMPARISON WITH USING TEXT PROMPTS

We've conducted experiments with InternVL2-4B/8B on VSI-Bench using the prompt "Please anchor common objects across frames for reference.", and the results are provided in Table 16. Explicitly prompting MLLMs brings marginal gains on VSI-Bench, while VideoAnchor improves significantly under the same settings.

Table 16: Comparison of VideoAnchor with using text prompts.

| Models | VSI-Bench Score |
|---|---|
| InternVL2-4B (8 frames) | 31.7 |
| +Prompt | 32.1 |
| +VideoAnchor | **34.5** |
| InternVL2-8B (8 frames) | 34.6 |
| +Prompt | 35.1 |
| +VideoAnchor | **37.8** |

## L    STATEMENT ON LLM USAGE

We acknowledge the use of large language models (LLMs) exclusively for writing assistance, limited to polishing grammar and style. No part of the research design, experiments, analysis, or core contributions relied on LLMs, and all text was critically reviewed and finalized by the authors.

