# OpenReview forum: "VideoAnchor: Reinforcing Subspace-Structured Visual Cues for Coherent Visual-Spatial Reasoning"
_ICLR.cc/2026/Conference — ICLR 2026 Poster_

### Official Review · Reviewer_G6Ba · 2025-10-29

**Soundness:** 2
**Presentation:** 2
**Contribution:** 2
**Rating:** 4
**Confidence:** 3

**Summary:**

This paper proposes VideoAnchor, a training-free module that improves visual–spatial reasoning in MLLMs by anchoring attention on shared visual cues across frames. It leverages sparse subspace clustering to identify consistent visual structures and regularizes attention at inference time. Evaluated on VSI-Bench, All-Angles-Bench, and Video-MME, it achieves limited performance gains without retraining.

**Strengths:**

1.VideoAnchor operates purely at inference time, making it easily applicable to existing MLLMs without fine-tuning.

2.Comprehensive analysis: Includes ablation studies and visualization to explain the mechanism’s effect on attention coherence.

3.This work establishes a connection between sparse subspace clustering and attention mechanisms, offering a principled way to preserve visual consistency across frames.

**Weaknesses:**

1. The method primarily targets pairwise spatial relations by anchoring one shared object. It remains unclear how this mechanism would generalize to broader scenarios such as navigation, counting, or reasoning over complex multi-object relations involving three or more entities.

2. The paper evaluates InternVL with 8-frame input and Qwen2.5-VL/LLaVA-Video with 16 frames on VSI-Bench, but does not justify this inconsistency. Moreover, since most videos in VSI-Bench and Video-MME last longer than 60s, sparse frame sampling may miss visual continuity, raising questions about how anchoring remains effective when frame overlap is minimal.

3. While VideoAnchor fits multi-view scenarios, evaluation across more multi-view benchmarks such as SPAR-Bench [1] would strengthen claims of generalizability. Also, models vary across benchmarks; evaluating the same model series would provide a clearer understanding of cross-task generalization.

4. Important inference details (e.g., temperature, rounds) are missing. It is unclear whether the observed gains are statistically significant or could be influenced by inference variance rather than the VideoAnchor mechanism itself.

5. Since VideoAnchor modifies attention internally, I am curious how explicit prompt-based anchoring performs—for instance, instructing the model to first identify an anchor object and then describe spatial relations.

[1] From Flatland to Space: Teaching Vision-Language Models to Perceive and Reason in 3D, 2025.

**Questions:**

Please refer to the **Weaknesses**

---

> ### Author Response · Authors · 2025-11-21
> **Q1-Q3**
>
> **Q1. The method primarily targets pairwise spatial relations by anchoring one shared object. It remains unclear how this mechanism would generalize to broader scenarios such as navigation, counting, or reasoning over complex multi-object relations involving three or more entities.**
>
> We appreciate the reviewer’s concern, but **VideoAnchor does not anchor "one shared object", nor is it designed exclusively for pairwise spatial relations**. The mechanism is fundamentally object-agnostic.
>
> VideoAnchor leverages sparse subspace clustering (SSC) to identify groups of tokens that exhibit shared visual structures across frames. **The subspaces discovered in VideoAnchor often correspond to multiple shared regions or objects across frames.** VideoAnchor reinforces these shared features across the entire visual scene based on the calculated sharing expression scores in the MLLM attention, rather than anchoring on only one object.
>
> **The sofa example in Fig. 1 is merely a visual illustration intended to show how such subspaces may correspond to consistent regions across frames.** It was not intended to suggest that VideoAnchor operates by selecting a single anchor object.
>
> Furthermore, VSI-Bench and All-Angles-Bench already contain tasks involving multi-object, multi-entity reasoning, including navigation (route planning), spatial relations, and counting. As shown in Table 1 and Table 2, VideoAnchor consistently improves MLLM performance across these tasks. For example, on VSI-Bench, InternVL2-8B (8 frames) gains **+16.2 on object counting** and **+6.2 on route planning**. On All-Angles-Bench, LLaVA-Video-72B gains **+7.4 on relative distance tasks** and InternVL2.5-8B gains **+3.6 on relative direction tasks**.
>
>
> **Q2. The paper evaluates InternVL with 8-frame input and Qwen2.5-VL/LLaVA-Video with 16 frames on VSI-Bench, but does not justify this inconsistency. Moreover, since most videos in VSI-Bench and Video-MME last longer than 60s, sparse frame sampling may miss visual continuity, raising questions about how anchoring remains effective when frame overlap is minimal.**
>
> We appreciate the reviewer’s concern. To clarify, though InternVL with 8-frame input and Qwen2.5-VL/LLaVA-Video with 16 frames on VSI-Bench are reported in Table 1, **we have provided the frame sampling robustness evaluation of VideoAnchor in Table 7**. VideoAnchor consistently improves performance across 8/16/32 frames for both InternVL2-8B and Qwen2.5-VL-7B, demonstrating robustness to frame sampling strategies.
>
> Regarding the concern that sparse sampling may yield minimal frame overlap and thus weaken the VideoAnchor's mechanism, we additionally evaluate VideoAnchor based on InternVL2-8B with 64 frames input on VSI-Bench, and the results are provided below, along with 8/16/32-frame results.
>
> | Frames | 8 | 16 | 32 | 64 |
> | ---- | ---- | ---- | ---- | ---- |
> | InternVL2-8B | 34.6 | 36.8 | 37.4 | 37.6 |
> | **+VideoAnchor** | 37.8 (+3.2) | 38.1 (+1.3) | 39.2 (+1.8) | 39.8 (+2.2) |
>
> The improvements are consistent across all settings, but not necessarily larger with more frames. We believe this behavior aligns with the nature of VSI-Bench: as an indoor-scene benchmark, scene geometry and object layouts remain relatively stable. Therefore, even with sparse sampling (e.g., 8 frames from long videos), the frames still maintain strong structural and visual continuity, allowing VideoAnchor to produce significant, stable gains even when frame overlap is minimal. We've included the content in the revised version.
>
>
>
> **Q3. While VideoAnchor fits multi-view scenarios, evaluation across more multi-view benchmarks such as SPAR-Bench would strengthen claims of generalizability. Also, models vary across benchmarks; evaluating the same model series would provide a clearer understanding of cross-task generalization.**
>
> We thank the reviewer's constructive comment. We've evaluated VideoAnchor with InterVL2-4B/8B and InterVL2.5-4B/8B on SPAR-Bench, and the results are shown below.
>
> | Models | Avg. |
> | ---- | ---- |
> | InternVL2-2B | 28.0 |
> | **+VideoAnchor** | **29.6 (+1.6)** |
> | InternVL2.5-2B | 30.0 |
> | **+VideoAnchor** | **31.2 (+1.2)** |
> | InternVL2-4B | 32.0 |
> | **+VideoAnchor** | **33.3 (+1.3)** |
> | InternVL2.5-4B | 30.5 |
> | **+VideoAnchor** | **33.1 (+2.6)** |
>
> VideoAnchor achieves consistent improvement over baseline MLLMs. Besides, VideoAnchor consistently improve various MLLMs on All-Angles-Bench in Table 2. We believe the results on multi-view sceneraio benchmarks (SPAR-Bench and All-Angles-Bench) validate the claim of generalizability.
>
> For the cross-task generalization with one same model series, **we've evaluated InternVL series on VSI-Bench/All-Angles-Bench/Video-MME/SPAR-Bench and LLaVA-Video series on VSI-Bench/All-Angles-Bench/Video-MME, and the performance gains are consistent**. Therefore, we believe the cross-task generalization of VideoAnchor is validated comprehensively. We've included the content in the revised version.

---

> ### Author Response · Authors · 2025-11-21
> **Q4 & Q5**
>
> **Q4. Important inference details (e.g., temperature, rounds) are missing. It is unclear whether the observed gains are statistically significant or could be influenced by inference variance rather than the VideoAnchor mechanism itself.**
>
> We appreciate the reviewer’s concern. In all experiments, we adopt **do_sample=False, num_beams=1** during inference. Under this setting, sampling-related parameters such as temperature and top-p are not used for generation and thus have no effect on the output. **Consequently, the models' responses are fully deterministic and consistent across runs, eliminating the possibility of inference variance.**
>
> The only difference brought by VideoAnchor is the attention regularization with the calculated $\gamma_{Q/K/V}$, ensuring that the reported improvements originate from the VideoAnchor's mechanism. We've clarified these inference details in the revised manuscript in Appendix.
>
>
>
> **Q5. Since VideoAnchor modifies attention internally, I am curious how explicit prompt-based anchoring performs—for instance, instructing the model to first identify an anchor object and then describe spatial relations.**
>
> We thank the reviewer's constructive comment. We've conducted experiments with InternVL2-4B/8B on VSI-Bench using the prompt **"Please anchor common objects across frames for reference."**, and the results are provided below.
>
> | Models | Avg. |
> | ---- | ---- |
> | InternVL2-4B (8 frames) | 31.7 |
> | +Prompt | 32.1 |
> | **+VideoAnchor** | **34.5** |
> | InternVL2-8B (8 frames) | 34.6 |
> | +Prompt | 35.1 |
> | **+VideoAnchor** | **37.8** |
>
> **Explicitly prompting MLLMs brings marginal gains on VSI-Bench, while VideoAnchor improves significantly under the same settings.** We've included these results in the revised manuscript.

---

> > ### Author Response · Authors · 2025-11-28
> > **Looking forward to further discussions!**
> >
> > Dear Reviewer G6Ba,
> >
> > Thank you for your insightful comments. We were wondering if our response and revision have resolved your concerns. We have attempted to address your initial questions through our replies and are eager to clarify any further points you might raise. Please feel free to provide additional feedback. We greatly appreciate your continued engagement.
> >
> > Best regards,
> > Authors

---

### Official Review · Reviewer_tF4U · 2025-10-31

**Soundness:** 3
**Presentation:** 3
**Contribution:** 3
**Rating:** 6
**Confidence:** 4

**Summary:**

This paper introduces VideoAnchor, a plug-and-play module designed to enhance visual–spatial reasoning in multimodal large language models (MLLMs) during inference. The key idea is to exploit the self-expressiveness property of Sparse Subspace Clustering (SSC) to uncover consistent visual structures across frames, and then translate these structural relations into token-wise scaling factors that modulate attention.

VideoAnchor consists of two components: 1. Subspace-to-Scaler Unit – Applies SSC to visual tokens extracted from multiple frames, computes each token’s “anchor strength” based on its intra-subspace connectivity, and generates scaling coefficients for the Q/K/V projections. 2. Attention Regularization Unit – Injects these scalers into the attention computation, amplifying stable visual regions and mitigating text-dominated biases without retraining. The method operates entirely at inference time, is model-agnostic, and can be seamlessly integrated into all self-attention layers of various MLLMs. Experiments across multi-frame and multi-view benchmarks demonstrate consistent accuracy gains (typically +1–3%).

**Strengths:**

1）Clear Conceptual Novelty. The paper introduces a distinctive idea by incorporating the self-expressiveness property of Sparse Subspace Clustering (SSC) into Transformer attention modulation. This provides a theoretically grounded and interpretable way to reinforce visual anchoring during inference.

2) Training-Free and Plug-and-Play Design. VideoAnchor operates entirely at inference time without parameter updates or retraining. Its plug-and-play nature makes it broadly compatible with various multimodal large language models (e.g., InternVL, LLaVA, Qwen-VL).

3) Strong Interpretability. The approach offers explicit geometric and semantic interpretability: subspace consistency identifies stable visual anchors, and attention amplification directly follows from these structural relations, unlike opaque learned gates.

4) Comprehensive Ablation Analysis. The paper includes systematic ablations on Q/K/V scaling, gating placement, and subspace cardinality, providing convincing evidence for the soundness of each design choice.

**Weaknesses:**

1）Lack of Quantitative Efficiency and Complexity Analysis. Although the paper emphasizes that VideoAnchor is lightweight, it provides no explicit measurements of runtime, FLOPs, or latency, leaving the actual computational efficiency unverified. While the ADMM optimization process is described, there is no formal analysis of its convergence behavior or computational scaling under realistic token counts.


2) No Discussion of Scalability to Long Videos or High-Resolution Inputs. Since the SSC step scales quadratically with the number of tokens, potential issues in long-sequence or high-resolution scenarios remain untested and unexplored.


3) Empirical Scope Limited to Spatial Reasoning Tasks. VideoAnchor appears to be a more general approach, but the evaluation focuses on visual–spatial reasoning; generalization to broader multimodal reasoning tasks (e.g., temporal dynamics, abstract understanding) remains unclear.

**Questions:**

1) Have the authors verified compatibility with high-performance inference frameworks (e.g., vLLM, FlashAttention, xFormers)? Does the multiplicative scaling require any modification to fused attention kernels or memory layouts?

2) The choice of subspace number and the scaling coefficients (α_Q, α_K, α_V) lacks principled derivation or sensitivity analysis beyond empirical tuning.

---

> ### Author Response · Authors · 2025-11-21
> **Q1-Q3**
>
> **Q1. Lack of Quantitative Efficiency and Complexity Analysis. Although the paper emphasizes that VideoAnchor is lightweight, it provides no explicit measurements of runtime, FLOPs, or latency, leaving the actual computational efficiency unverified. While the ADMM optimization process is described, there is no formal analysis of its convergence behavior or computational scaling under realistic token counts.**
>
> Thanks for your review! We acknowledge that the iterative ADMM optimization in SSC introduces additional latency and additional complexity ($O(N^2)$ with the number of tokens $N$) compared to the baseline. To mitigate this, we implemented the ADMM solver using CuPy for efficient GPU acceleration, which significantly reduces the computational bottleneck. Furthermore, we notice that VideoAnchor is robust to the ADMM convergence threshold ($\epsilon$), thus choosing suitable relaxed thresholds reduces runtime with marginal impact on the performance. Measured on NVIDIA V100 GPUs, the optimized VideoAnchor increases inference latency by only 0.8s per sample while yielding a substantial +3.2 gain on VSI-Bench (as shown in the table below), demonstrating a highly acceptable trade-off between efficiency and accuracy. **Importantly, we emphasize that the core contribution of this work lies in unveiling the novel insight that subspace self-expressiveness effectively enhances visual-spatial reasoning, rather than designing a specialized low-latency clustering algorithm.** We've included the content in the revised version.
>
> | Models | Runtime (s/iter) | VSI-Bench Score |
> | ---- | ---- | ---- |
> |InternVL2-8B (8 frames)|6.0|34.6|
> |**+ VideoAnchor**|6.8|37.8 (+3.2)|
>
> **Q2. No Discussion of Scalability to Long Videos or High-Resolution Inputs. Since the SSC step scales quadratically with the number of tokens, potential issues in long-sequence or high-resolution scenarios remain untested and unexplored.**
>
> Thanks for your constructive comment! The SSC actually has the complexity $O(N^2)$ with the number of tokens $N$. We additionally evaluate VideoAnchor based on InternVL2-8B with 64 frames input on VSI-Bench, and the results are provided below, along with 8/16/32-frame results. This improvement is comparable to, and even higher than, the gains observed at 16 or 32 frames (+1.3 and +1.8, respectively). These empirical results confirm that VideoAnchor effectively scales to long videos. We've included the content in the revised version.
>
> | Frames | 8 | 16 | 32 | 64 |
> | ---- | ---- | ---- | ---- | ---- |
> | InternVL2-8B | 34.6 | 36.8 | 37.4 | 37.6 |
> | **+VideoAnchor** | 37.8 (+3.2) | 38.1 (+1.3) | 39.2 (+1.8) | 39.8 (+2.2) |
>
> **Q3. Empirical Scope Limited to Spatial Reasoning Tasks. VideoAnchor appears to be a more general approach, but the evaluation focuses on visual–spatial reasoning; generalization to broader multimodal reasoning tasks (e.g., temporal dynamics, abstract understanding) remains unclear.**
>
> Thanks for your review! We've evaluated VideoAnchor on a general video understanding benchmark, Video-MME, in Table 10 in Appendix. VideoAnchor consistently improves performance across all evaluated models, with gains ranging from +0.3 to +1.2. The improvements are especially notable on spatial-related tasks (+1.8 ~ +4.6) in Table 3 (right). **This aligns with our discussion in Section 3.4 that SSC functions by "encoding geometric dependencies and higher-level object continuity."** While these structural cues provide a foundational benefit for general video understanding—such as maintaining object consistency across temporal dynamics—their impact is naturally maximized in tasks demanding explicit visual-spatial reasoning.

---

> ### Author Response · Authors · 2025-11-21
> **Q4 & Q5**
>
> **Q4. Have the authors verified compatibility with high-performance inference frameworks (e.g., vLLM, FlashAttention, xFormers)? Does the multiplicative scaling require any modification to fused attention kernels or memory layouts?**
>
> Thank you for providing this constructive comment! Our VideoAnchor mechanism introduces a multiplicative re-weighting of attention, which can be equivalently expressed as an additive log-bias added before the softmax. In principle, such a bias could be injected into FlashAttention by converting it into a dense attention_mask. However, this approach leads to prohibitively high memory consumption, especially for long video sequences.
>
> To address this, we refer to a recent NeurIPS 2025 paper FlashBias [1] (many thanks!), a high-performance inference framework designed to integrate prior-driven attention biases compatible with efficient backends like Triton or PyTorch's SDPA. Given a modified attention $${O}\=softmax(\frac{{Q}{K}^\top}{d_h}+{B}){V}$$, when the attention bias is low-rank and can be decomposed as $B=Q_{bias}{K}_{bias}^\top$, FlashBias provides an efficient implementation—without any dense mask construction—and the inference cost becomes very close to pure FlashAttention with no extra memory overhead.
>
> The introduced attention_bias in VideoAnchor is $log({\gamma}_Q{\gamma}_K^\top)$ (provided in Eq. (7)), which can be exactly decomposed with $rank\=2$ by:
>
> **$$\log({{\gamma}_Q}{{\gamma}_K^\top})=(\log{\gamma}_Q,1\_{{n_Q}\times1})(1\_{{n_K}\times1},\log{\gamma_K})^\top.$$**
>
> This allows us to directly adopt the FlashBias kernel and guarantees: no need for dense masking, no extra memory overhead, and the inference efficiency that could match standard FlashAttention. Thus, VideoAnchor naturally fits into the class of attention-bias mechanisms supported efficiently by FlashBias, providing both theoretical correctness and practical compatibility with modern high-performance inference frameworks. With FlashBias, we test VideoAnchor's inference efficiency based on Qwen2.5-VL-7B (16 frames) on VSI-Bench using NVIDIA 80G A100 GPUs. Note that for this kernel-level comparison, the SSC clustering results were pre-cached. The runtime, GPU memory allocation, and the results are provided in the table below. VideoAnchor incurs negligible latency increase and maintains the GPU memory consumption compared to the baseline, validating that attention regularization units in VideoAnchor are compatible with high-performance inference frameworks. We've included the content in the revised version in Appendix.
>
> | Models | Runtime (s/iter) |GPU Memory (GB)| VSI-Bench Score |
> | ---- | ---- | ---- | ---- |
> |Qwen2.5-VL-7B (16 frames, FlashAttention)|5.8|38.7|31.8|
> |+ VideoAnchor (16 frames, FlashBias-SDPA)|6.1|38.7|33.3 (+1.5)|
>
> [1] Wu, Haixu, et al. "FlashBias: Fast Computation of Attention with Bias." NeurIPS 2025.
>
> **Q5. The choice of subspace number and the scaling coefficients (α_Q, α_K, α_V) lacks principled derivation or sensitivity analysis beyond empirical tuning.**
>
> Thanks for your constructive review! To further validate the robustness in $\alpha_{Q/K/V}$ setting of VideoAnchor, we conduct experiments with InternVL2-4B (8 frames) on VSI-Bench. As detailed in the table below, we varied $\alpha_Q$, $\alpha_K$, and $\alpha_V$ with a step size of 0.2 within the ranges of $[1.5, 3.5]$, $[1.0, 3.0]$, and $[2.0, 4.0]$, respectively. The results suggest that the model performance remains relatively consistent across these variations, with scores fluctuating within the range of 33.6 to 34.5. Furthermore, these configurations consistently show improvements over the baseline score of 31.7, implying that VideoAnchor may not be overly sensitive to precise hyperparameter tuning within these intervals. We've included the content in the revised version in Appendix.
>
> | | Values | VSI-Bench Score |
> | ---- | ---- | ---- |
> |Baseline|-|31.7|
> |$\alpha_Q$|1.5-3.5|34.3-34.5|
> |$\alpha_K$|1.0-3.0|34.0-34.5|
> |$\alpha_V$|2.0-4.0|33.6-34.5|

---

> > ### Comment · Reviewer_tF4U · 2025-11-28
> >
> > Thank you for the additional experiments addressing efficiency and generalization.
> > My concerns have been resolved.
> > I maintain my original score.

---

> > > ### Author Response · Authors · 2025-11-28
> > > **Thank you again for you comments!**
> > >
> > > We greatly appreciate your feedback of our paper. Thank you again for your valuable time and input!

---

### Official Review · Reviewer_84JG · 2025-11-01

**Soundness:** 3
**Presentation:** 3
**Contribution:** 3
**Rating:** 4
**Confidence:** 4

**Summary:**

This paper introduces *VideoAnchor*, a plug-and-play attention modulation module designed for test-time enhancement. The core idea is to cluster multi-frame or multi-view visual tokens into semantic subspaces using sparse subspace clustering, compute a *shared expression score* for each token, and convert it into scaling factors applied to QKV in the attention mechanism.  Extensive experiments on VSI-Bench, All-Angles-Bench, and Video-MME demonstrate consistent improvements across multiple multimodal LLM backbones, supported by ablation studies and visual analyses.

**Strengths:**

1. A training-free and model-agnostic *subspace-to-scaler* mechanism that can be universally applied across architectures, linking self-expressive subspace modeling with transformer attention in a novel way.
2. Comprehensive experimental validation on diverse benchmarks with consistent gains and detailed ablations covering QKV scaling, clustering methods, and subspace counts.

**Weaknesses:**

1. The comparison with other training-free enhancement approaches (e.g., DC2’s high-resolution perception [1], ControlMLLM’s visual prompting [2], VisionFuse’s multi-encoder fusion [3]) lacks equal experimental settings and quantitative cost-performance analysis.
2. Missing a deeper comparison or hybridization with learning-based reinforcement approaches such as Video-R1 or T-GRPO [4], the paper only briefly mentions them in the appendix without unified benchmarks.
3. The failure cases and generalization limits (e.g., occlusions, fast viewpoint changes, low-texture regions) are underexplored, with only anecdotal visual evidence.
4. The SSC plus spectral clustering steps could introduce high latency when processing long videos or thousands of tokens, yet no systematic runtime or throughput evaluation is provided.
5. Hyperparameter sensitivity is superficially treated. Although claimed “robust,” only fixed αQ/αK/αV and subspace counts are reported, without cross-dataset robustness or automatic selection strategies.

[1] Li et al. *DC2: Decoupled Cross-resolution Conditioning for High-Resolution Multimodal Models*.

[2] Hu et al. *VisionFuse: Multi-Encoder Fusion for Multimodal Reasoning*.

[3] Xu et al. *ControlMLLM: Test-time Control for Multimodal Large Language Models*.

[4] Wang et al. *Video-R1: Reinforcing Video Reasoning in Multimodal Large Language Models*.

**Questions:**

1. What is the computational complexity of VideoAnchor in terms of frames, tokens, and subspace counts? A runtime comparison with DC2 and VisionFuse would clarify its practicality.
2. Does the sharing expression score bias attention toward large clusters, potentially overlooking small but semantically critical objects? Could weighting or normalization mitigate this issue?
3. When strong appearance changes or motion blur occur, how does the linear self-expression assumption of SSC hold? Have the authors considered nonlinear or graph-based subspace alternatives?
4. Could VideoAnchor be combined with learning-based methods such as Video-R1 or GRPO fine-tuning to further improve robustness and visual grounding?
5. Across benchmarks,VSI-Bench, All-Angles-Bench, and Video-MME,which specific sub-tasks (e.g., spatial relation, counting, manipulation) benefit most, and does the improvement mainly come from enhanced perception or reasoning?

---

> ### Author Response · Authors · 2025-11-21
> **Q1 & Q2**
>
> Thanks for your review. **We kindly remind you that the four references you provided are not consistent with the truth.**
>
> [1] The listed “Li et al. DC2: Decoupled Cross-resolution Conditioning for High-Resolution Multimodal Models” Should be “Wang, et al. Divide, conquer and combine: A training-free framework for high-resolution image perception in multimodal large language models. AAAI 2025”.
> [2] The listed “Hu et al. VisionFuse: Multi-Encoder Fusion for Multimodal Reasoning” should be “Chen, et al. Enhancing perception capabilities of multimodal llms with training-free fusion. arXiv”.
> [3] The listed “Xu et al. ControlMLLM: Test-time Control for Multimodal Large Language Models” should be “Wu, et al. Controlmllm: Training-free visual prompt learning for multimodal large language models. NeurIPS 2024”.
> [4] The listed “Wang et al. Video-R1: Reinforcing Video Reasoning in Multimodal Large Language Models” should be “Feng, et al. Video-r1: Reinforcing video reasoning in mllms. arXiv”.
> The listed [1][2][3] contain incorrect titles and incorrect authors, and the listed [4] contains incorrect authors.
>
> **Q1. The comparison with other training-free enhancement approaches (e.g., DC2’s high-resolution perception, ControlMLLM’s visual prompting, VisionFuse’s multi-encoder fusion) lacks equal experimental settings and quantitative cost-performance analysis.**
>
> Thanks for your review. Although the citations in the review are factually incorrect (as noted in the General Response), we address these methods based on the abbreviations in our paper. Comparisons are infeasible due to fundamental mismatches:
>
> 1) DC2 targets high-resolution image perception, whereas VideoAnchor is explicitly designed for video understanding (spatial reasoning).
> 2) ControlMLLM relies on manual user prompts (e.g., bounding boxes), while VideoAnchor is fully automatic and unsupervised.
> 3) VisionFuse requires multiple visual encoders (e.g., CLIP + DINO), which is architecturally incompatible with our standard single-encoder setting.
>
> **Q2. Missing a deeper comparison or hybridization with learning-based reinforcement approaches such as Video-R1 or T-GRPO, the paper only briefly mentions them in the appendix without unified benchmarks.**
>
> The comparison with Video-R1 in Appendix validates the better robustness of frame sampling of VideoAnchor than Video-R1. Additionally, we further develop a Trainable VideoAnchor (T-VideoAnchor), which can be trained on supervised datasets for improved visual-spatial reasoning. We expand the channels of $\alpha_{Q/K/V}$ to align the number of attention heads and make them learnable within each decoder layer in MLLMs, forming a lightweight parameter-efficient fine-tuning (PEFT) module. Motivated by the finding that SSC-derived representativeness inherently encodes geometric dependencies and object continuity, this design allows the model to adaptively calibrate the enhancement magnitude applied to these semantically coherent and structurally central elements via layer-wise and head-specific modulation. Furthermore, we incorporate low-rank adaptation (LoRA) to optimize general representations alongside T-VideoAnchor, resulting in improved visual-spatial reasoning.
>
> We perform supervised fine-tuning on the VSI-Train-10k dataset for 3 epochs. Specifically, we employ LoRA with a rank of 8. As shown in the table below, our proposed T-VideoAnchor significantly outperforms GRPO-trained Video-R1 (+17.4), achieving superior spatial reasoning capabilities. We've included the content in the revised version.
>
> | Models | Tranable Params. | Training Datasets | VSI-Bench Score |
> | ---- | ---- | ---- | ---- |
> |Qwen2.5-VL-7B|-|-|31.8|
> | Video-R1 | Unknown | Video-R1-CoT-165k \& Video-R1-260k | 34.6 (+2.8) |
> | Qwen2.5-VL-7B + LoRA (rank=8) | 20.19M | VSI-Train-10k | 47.7 (+15.9) |
> | **Qwen2.5-VL-7B + LoRA (rank=8) + T-VideoAnchor** | 20.19M | VSI-Train-10k | **52.0 (+20.2)** |

---

> ### Author Response · Authors · 2025-11-21
> **Q3-Q5**
>
> **Q3. The failure cases and generalization limits (e.g., occlusions, fast viewpoint changes, low-texture regions) are underexplored, with only anecdotal visual evidence.**
>
> Thanks for your comment! We've explored the failure cases of VideoAnchor with InterVL2-8B (8 frames) on VSI-Bench and provided two main categories in Section M in Appendix. In failure case 1, as shown in Fig. 7, the model may exhibit counting ambiguities; this stems from the intrinsic mechanism of Sparse Subspace Clustering (SSC), which tends to merge highly correlated visual features into a single subspace, thereby blurring the attention boundaries between distinct but near-identical instances. In failure case 2, as shown in Fig. 8, the model struggles with questions regarding fleeting events that occur between sampled frames. This limitation is attributed to the discrete nature of temporal sampling rather than the clustering module itself, as VideoAnchor functions as a feature enhancement mechanism and cannot recover visual information physically absent from the input grid. We've included the content in the revised version.
>
> **Q4. The SSC plus spectral clustering steps could introduce high latency when processing long videos or thousands of tokens, yet no systematic runtime or throughput evaluation is provided.**
>
> Thanks for your review! We acknowledge that the iterative ADMM optimization in SSC introduces additional latency and additional complexity ($O(N^2)$ with the number of tokens $N$) compared to the baseline. To mitigate this, we implemented the ADMM solver using CuPy for efficient GPU acceleration, which significantly reduces the computational bottleneck. Furthermore, we notice that VideoAnchor is robust to the ADMM convergence threshold ($\epsilon$), thus choosing suitable relaxed thresholds reduces runtime with marginal impact on the performance. Measured on NVIDIA V100 GPUs, the optimized VideoAnchor increases inference latency by only 0.8s per sample while yielding a substantial +3.2 gain on VSI-Bench (as shown in the table below), demonstrating a highly acceptable trade-off between efficiency and accuracy. **Importantly, we emphasize that the core contribution of this work lies in unveiling the novel insight that subspace self-expressiveness effectively enhances visual-spatial reasoning, rather than designing a specialized low-latency clustering algorithm.** We've included the content in the revised version.
>
> | Models | Runtime (s/iter) | VSI-Bench Score |
> | ---- | ---- | ---- |
> |InternVL2-8B (8 frames)|6.0|34.6|
> |**+ VideoAnchor**|6.8|37.8 (+3.2)|
>
> **Q5. Hyperparameter sensitivity is superficially treated. Although claimed “robust,” only fixed αQ/αK/αV and subspace counts are reported, without cross-dataset robustness or automatic selection strategies.**
>
> Thanks for your comment! To further validate the robustness in $\alpha_{Q/K/V}$ setting of VideoAnchor, we conduct experiments with InternVL2-4B (8 frames) on VSI-Bench. As detailed in the table below, we varied $\alpha_Q$, $\alpha_K$, and $\alpha_V$ with a step size of 0.2 within the ranges of $[1.5, 3.5]$, $[1.0, 3.0]$, and $[2.0, 4.0]$, respectively. The results suggest that the model performance remains relatively consistent across these variations, with scores fluctuating within the range of 33.6 to 34.5. Furthermore, these configurations consistently show improvements over the baseline score of 31.7, implying that VideoAnchor may not be overly sensitive to precise hyperparameter tuning within these intervals. We've include the content in the revised version in Appendix.
>
> | | Values | VSI-Bench Score |
> | ---- | ---- | ---- |
> |Baseline|-|31.7|
> |$\alpha_Q$|1.5-3.5|34.3-34.5|
> |$\alpha_K$|1.0-3.0|34.0-34.5|
> |$\alpha_V$|2.0-4.0|33.6-34.5|

---

> ### Author Response · Authors · 2025-11-21
> **Q6-Q9**
>
> **Q6. Does the sharing expression score bias attention toward large clusters, potentially overlooking small but semantically critical objects? Could weighting or normalization mitigate this issue?**
>
> Thank you for this insightful observation. We clarify that our proposed self-expression score is calculated as the product of both the cluster size and the reconstruction coefficient. This design effectively balances the trade-off between noise and semantics:
>
> 1) Noise Suppression: Weighting by cluster size is intended to suppress visual noise, which usually appear as very small, fragmented, and incoherent clusters in unsupervised settings.
> 2) Preserving Critical Objects: Notably, semantically critical small objects typically exhibit exceptionally strong self-expression (high internal structural coherence) compared to random noise. This high reconstruction coefficient effectively compensates for their smaller cluster size, ensuring that semantically critical small objects remain highlighted in the attention map.
>
> In summary, our proposed self-expression score does not overlook small but critical objects, and it actually performs like a selective noise filter in some way. And based on the score, VideoAnchor can enhance critical and coherent visual cues within the input videos or multiple images.
>
> **Q7. When strong appearance changes or motion blur occur, how does the linear self-expression assumption of SSC hold? Have the authors considered nonlinear or graph-based subspace alternatives?**
>
> Thanks for your comment. While raw pixels exhibit nonlinearity under strong changes, VideoAnchor operates on the high-dimensional deep feature space of the visual encoder. Deep networks effectively map complex visual variations (e.g., lighting, deformation) into flattened, linearly separable semantic manifolds, making the linear self-expression assumption valid for semantic clustering. The optimization objective of SSC is inherently robust to sparse errors, treating motion blur or transient artifacts as noise/outliers to be suppressed rather than reconstructed.
>
> We did consider nonlinear alternatives (e.g., Kernel SSC). However, we prioritized standard SSC to maintain a lightweight, plug-and-play design, as nonlinear methods introduce significant computational overhead that hinders efficient test-time inference.
>
> **Q8. Could VideoAnchor be combined with learning-based methods such as Video-R1 or GRPO fine-tuning to further improve robustness and visual grounding?**
>
> Thanks for your comment. We've evaluated Video-R1 (16 frames) with VideoAnchor on VSI-Bench and results are provided below. VideoAnchor improves Video-R1 by 1.6, validating the enhanced visual spatial reasoning of VideoAnchor based on learning-based or GRPO fine-tuned MLLMs.
>
> | Models | VSI-Bench Score |
> | ---- | ---- |
> |Video-R1|34.6|
> |**Video-R1 + VideoAnchor**|**36.2 (+1.6)**|
>
> **Q9. Across benchmarks,VSI-Bench, All-Angles-Bench, and Video-MME,which specific sub-tasks (e.g., spatial relation, counting, manipulation) benefit most, and does the improvement mainly come from enhanced perception or reasoning?**
>
> Thanks for your comment! VideoAnchor performs significantly better on spatial-related tasks. **This aligns with our discussion in Section 3.4 that SSC functions by "encoding geometric dependencies and higher-level object continuity."** While these structural cues provide a foundational benefit for general video understanding—such as maintaining object consistency across temporal dynamics—their impact is naturally maximized in tasks demanding explicit visual-spatial reasoning. Thus, regarding the source of improvement, we conclude that the gains primarily stem from enhanced perception—specifically, the robust grounding of visual evidence.

---

> > ### Comment · Reviewer_84JG · 2025-11-21
> > **Response to the Authors’ Rebuttal**
> >
> > Thank you for the authors’ responses. The updated review already provided the correct paper titles rather than the initial placeholder summaries, along with links to the original works. The authors should have verified the latest version of the review before submitting the rebuttal.
> >
> > After reading the rebuttal, I acknowledge that some of my initial concerns have been partially addressed. However, several critical issues remain unresolved. For example, there are apparent fairness concerns in the experimental setup. In Q2, the authors introduce a new trainable variant, T-VideoAnchor, in order to compare against Video-R1. This raises a fundamental question: is the contribution of this paper meant to be a test-time method or a trainable approach? Moreover, T-VideoAnchor is trained on 10k samples, can such a comparison be considered fair?
> >
> > In addition, many of the authors’ answers remain purely explanatory rather than empirically validated. In Q1, the authors state that comparisons with other training-free methods are infeasible due to task differences. This is inadequate. In practice, all these works aim to enhance visual perception to strengthen reasoning. Even if the tasks differ, it is still possible to provide cost–benefit curves under unified input conditions, sub-task comparisons, or controlled ablations. Similarly, in Q9, the authors could have offered sub-task-level ablation studies, in Q6, the claim that small objects exhibit high reconstruction coefficients that compensate for small cluster sizes is purely theoretical and lacks experimental support. The same issue also appears in Q7 and elsewhere.
> >
> > In summary, this work still exhibits notable limitations and has room for improvement. I will therefore maintain my negative score.

---

> > > ### Author Response · Authors · 2025-11-25
> > > **Response to Reviewer 84JG**
> > >
> > > Dear Reviewer,
> > >
> > > Thank you for your response.
> > >
> > > We would like to clarify that our rebuttal was prepared and submitted based on the original review. We noticed that you updated the references—correcting the paper titles and removing the incorrect author names—several hours before our rebuttal submission. We acknowledge these corrections.
> > >
> > > T-VideoAnchor was initially designed for the discussion with Reviewer H6LV. In my newly submitted version, to minimize potential confusion, T-VideoAnchor has been removed, and the core contribution is enhancing visual-spatial reasoning with SSC at inference time.
> > >
> > > Regarding your mention of DC2, ControlMLLM, and VisionFuse, we reemphasize here that these three methods are designed for **image understanding**, which is very different from **video understanding**, for which the VideoAnchor is proposed, and applying them to video inputs may lead to catastrophic failure.
> > >
> > > Authors

---

### Official Review · Reviewer_H6LV · 2025-11-05

**Soundness:** 3
**Presentation:** 3
**Contribution:** 3
**Rating:** 6
**Confidence:** 3

**Summary:**

The paper considers the visual-spatial reasoning in the context of MLLMs, and proposes VideoAnchor, a plug-and-play module, to address the limitation of current MLLMs that they lack a mechanism to consistently preserve visual cues cross frames. Specifically, the paper aims to draw connection between the self-expressive property in sparse subspace clustering (SSC) and attention mechanisms in Transformers. To implement this, the proposed VideoAnchor consists of two units: (1) subspace-to-scaler unit, and (2) attention regularization unit. Methods and empirical evaluations are presented.

**Strengths:**

Overall, the paper is easy to follow. The problem setting, the intuition/motivation (that token organization in the semantic subspace helps), the technical approach, and the empirical evaluations are presented in a relatively clear and organized manner. The utilization and illustration of SSC (e.g., the discussions in Section 3.4, illustrations in Section 4.2) provide different perspectives and justify the incorporation of two units in VideoAnchor. Experimental results also show improvements.

**Weaknesses:**

The improvement over baseline is relatively modest. This is not entirely unexpected now that the module does not require re-training. That being said, I am wondering if author(s) can share thoughts on whether the plug-and-play module can have a more "integrated" version, in the sense that how would VideoAnchor operate if (lightweight) retraining is feasible. Additional discussion on the tradeoff between expected performance gain and the retraining effort may be preferable (e.g., to better justify the contribution), in addition to the claim that VideoAnchor is plug-and-play.

**Questions:**

(As mentioned in "Weakness" Section) Can author(s) provide further discussion/justification on the tradeoff between expected performance gain and the retraining effort?

---

> ### Author Response · Authors · 2025-11-21
> **Q1. Can author(s) provide further discussion/justification on the tradeoff between expected performance gain and the retraining effort?**
>
> **Q1. Can author(s) provide further discussion/justification on the tradeoff between expected performance gain and the retraining effort?**
>
> We appreciate the reviewer’s very constructive reviews!
>
> To validate it, we further develop a Trainable VideoAnchor (T-VideoAnchor), which can be trained on supervised datasets for improved visual-spatial reasoning. We expand the channels of $\alpha_{Q/K/V}$ to align the number of attention heads and make them learnable within each decoder layer in MLLMs, forming a lightweight parameter-efficient fine-tuning (PEFT) module. Motivated by the finding that SSC-derived representativeness inherently encodes geometric dependencies and object continuity, this design allows the model to adaptively calibrate the enhancement magnitude applied to these semantically coherent and structurally central elements via layer-wise and head-specific modulation. Furthermore, we incorporate low-rank adaptation (LoRA) to optimize general representations alongside T-VideoAnchor, resulting in improved visual-spatial reasoning.
>
> We perform supervised fine-tuning on the VSI-Train-10k [1] dataset for 3 epochs based on the framework of LLaMA-Factory [2]. Specifically, we employ LoRA with a rank of 8. As shown in the table below, while standard LoRA fine-tuning already yields a substantial improvement over the baseline (47.7 vs. 31.8), incorporating T-VideoAnchor further elevates the performance to 52.0, with significant improvements over baseline (+20.2) and LoRA fine-tuning (+4.3), while maintaining only 20.19M trainable parameters. Notably, T-VideoAnchor significantly outperforms GRPO-trained Video-R1 (+17.4), achieving superior spatial reasoning capabilities. The training costs only 5-6 hours with 8 A100 GPUs.
>
> | Models | Tranable Params. | Training Datasets | VSI-Bench Score |
> | ---- | ---- | ---- | ---- |
> |Qwen2.5-VL-7B|-|-|31.8|
> | Video-R1 | Unknown | Video-R1-CoT-165k \& Video-R1-260k | 34.6 (+2.8) |
> | Qwen2.5-VL-7B + LoRA (rank=8) | 20.19M | VSI-Train-10k | 47.7 (+15.9) |
> | **Qwen2.5-VL-7B + LoRA (rank=8) + T-VideoAnchor** | 20.19M | VSI-Train-10k | **52.0 (+20.2)** |
>
> [1] Brown, Ellis, et al. "Benchmark Designers Should" Train on the Test Set" to Expose Exploitable Non-Visual Shortcuts." arXiv preprint arXiv:2511.04655 (2025).
>
> [2] Zheng, Yaowei, et al. "LlamaFactory: Unified Efficient Fine-Tuning of 100+ Language Models." Proceedings of the 62nd Annual Meeting of the Association for Computational Linguistics (Volume 3: System Demonstrations). 2024.

---

> ### Author Response · Authors · 2025-11-28
> **Looking forward to further discussions!**
>
> Dear Reviewer H6LV,
>
> Thank you for your insightful comments. We were wondering if our response and revision have resolved your concerns. We have attempted to address your initial questions through our replies and are eager to clarify any further points you might raise. Please feel free to provide additional feedback. We greatly appreciate your continued engagement.
>
> Best regards,
> Authors

---

### Author Response · Authors · 2025-11-21
**Grateful thanks to all reviewers**

We express our gratitude to all the reviewers for their comprehensive reviews and valuable feedback on our submission. The insightful reviews and comments will greatly motivate us to improve our work.

We are pleased to receive positive comments and support from all reviewers. Reviewer H6LV: VideoAnchor enhances visual-spatial reasoning from a different perspective by bridging SSC with attention in MLLMs. Reviewer 84JG: A training-free and model-agnostic subspace-to-scaler mechanism that can be universally applied across architectures, linking self-expressive subspace modeling with transformer attention in a novel way. Reviewer tF4U: Clear conceptual novelty, training-free and plug-and-play design, strong interpretability, and comprehensive ablation analysis. Reviewer G6Ba: This work establishes a connection between sparse subspace clustering and attention mechanisms, offering a principled way to preserve visual consistency across frames.

Their concerns are addressed point-to-point below. Thanks again for the review!

---

### Meta-Review · Area_Chair_tER2 · 2026-01-06

**Summary:**

The paper introduces a training-free, plug-and-play module that can be incorporated into modern multimodal LLMs to improve their spatial reasoning capabilities. Overall, the reviewers recognize the novelty of the design and appreciate the empirical performance gains. The major concerns were (i) missing experimental details and (ii) insufficient validation across models and datasets. During the discussion period, the authors provided extensive additional experiments, and after careful review, the AC believes that these results address most of the reviewers’ concerns. The AC therefore recommends acceptance and urges the authors to incorporate the reviewers’ feedback into the final version.

**Reviewer Concerns:**

See above.

**Reviewer Scores:**

See above.

---

### Decision · Program_Chairs · 2026-01-26

Accept (Poster)